# VIMPAC: VIDEO PRE-TRAINING VIA MASKED TOKEN PREDICTION AND CONTRASTIVE LEARNING

## ABSTRACT

Video understanding relies on perceiving the overall global content and modeling its internal connections (e.g., causality, movement, and spatio-temporal correspondence). To learn these interactions, we apply a mask-then-predict pre-training task on the discretized video tokens generated via VQ-VAE. Unlike language, where the text tokens are more independent, neighboring video tokens typically have strong correlations (e.g., consecutive video frames usually look similar), and hence uniformly masking individual tokens will make the task too trivial to learn useful representations. To deal with this issue, we propose a block masking strategy where we mask neighboring video tokens in both spatial and temporal domains. We also add a contrastive learning objective to further capture the global content by predicting whether the video clips are sampled from the same video. We pre-train our model on uncurated videos and show that our pre-trained model can reach state-of-the-art results on several video understanding datasets (e.g., SSV2, Diving48). Lastly, we provide detailed analyses of the model scalability and pre-training method design.

## 1 INTRODUCTION

In recent years, state-of-the-art self-supervised methods have been exploring different directions for pre-training images and text representations, with Contrastive Learning (CL) providing strong results for vision representation learning (Oord et al., 2018; Chen et al., 2020b; He et al., 2020; Chen et al., 2020c; Tian et al., 2020), and Language Modeling (LM) becoming the de-facto standard in language pre-training (Devlin et al., 2019; Liu et al., 2019; Yang et al., 2019; Lan et al., 2019). Both approaches are quite different from each other. A contrastive objective compares positive/negative examples at a coarse/sample level, focusing on global-content (e.g., for image classification) while a token modeling objective predict missing tokens from context at a much finer/sub-sample level to model sequential and short range interactions between tokens (e.g. in text generation tasks). Interestingly, video understanding naturally combines both types of requirements. 2D processing along the spatial dimensions of the video bears similarity to image processing, while 1D processing along the temporal dimension often involves modeling sequential events and short range coherence.

Hence, in this work, we propose to combine both text and image representation learning approaches for improved video pre-training, taking advantage of recent advances in self-supervised methods of both fields. We name our method as VIMPAC: VIdeo pre-training via Masked token Prediction And Contrastive learning. From language research, we adopt a 'masked language model' pre-training objective (Devlin et al., 2019) where a model is trained to reconstruct local masked regions in videos. From the computer vision world, we borrow a contrastive learning objective, specifically the InfoNCE (Oord et al., 2018) objective is applied on positive/negative video samples. While the masked language model objective encourages models to learn low-level semantics and sequential interaction, the contrastive loss provide a supervision for the model to learn more global and separable representations that are useful for many downstream tasks (e.g., action classification (Soomro et al., 2012b; Carreira & Zisserman, 2017)). The two objectives provide complementary signals for training: while short range correlations can be predominantly modeled from the training signal of the mask-and-predict task, the contrastive learning objective can provide signals on a more coarse-grained global-context and semantic level.

However, unlike language which is composed of discrete tokens from a compact vocabulary, videos are typically represented as RGB pixels in an almost continuous, high dimensional vector space. Naively masking pixels in videos induces a prohibitive computation cost while also tending to over-emphasize local details. To overcome these issues, we first tokenize input videos using the latent codes of a pretrained Vector Quantized-Variational Auto-Encoder (VQ-VAE) (van den Oord et al., 2017; Ramesh et al., 2021) to encode them in smaller quantized representations on which a reconstruction model can then be trained with a masked token modeling objective. In practice, we also discovered that models trained with a uniform random token masking strategy can fail to learn meaningful and useful visual representations as neighboring pixels may contain similar and correlated content (in particular along the temporal dimension), making the task of predicting a randomly masked token from its visible neighbors trivial. We therefore also introduce a block-masking scheme that simultaneously masking video tokens in a 3D spatio-temporal block. Reconstructing such an extended spatio-temporal cube requires performing long-range predictions, forcing the models to learn a more complex set of relations between the video tokens, resulting in better visual representations.

Our contrastive learning approaches also departs from previous work in several aspects. First, since we apply the contrastive objective on token-discretized video samples and in combination with the token modeling loss, we observe strong performance without requiring the usual extensive set of data augmentations (Chen et al., 2020b;c; Qian et al., 2021; Feichtenhofer et al., 2021). Second, we are able to leverage positive clip pairs that are temporally distant from each other (can be as far as 400 seconds away), while previous work favors using positives within a shorter range (maximum 36 seconds for uncurated videos in Feichtenhofer et al. (2021) or 10 seconds in Qian et al. (2021)).

We evaluate the performances of our method VIMPAC on several video understanding datasets, including two temporally-heavy tasks, SSV2 and Diving48 on which it achieves state-of-the-art results with regard to both self-supervised and supervised pre-training works, and a set of more spatially-heavy datasets, UCF101, HMDB51, and Kinetics-400, on which it also achieves competitive results. Overall, taking advantage of VQ-VAE discretized video tokens, we present a method for self-supervised learning of video representations that combines two general streams of research in self-supervision: masked language modeling and contrastive learning. Our contribution is 3-folds: ($i$) We apply the mask-then-predict task to video understanding and introduce the use of block masking. ($ii$) We propose a contrastive learning method which is able to achieve strong performance without spatial data augmentation. ($iii$) We empirically show that this method achieves strong performance on several video classification datasets, especially on temporally-heavy datasets, SSV2 and Diving48, where it sets new state-of-the-art results. We also present comprehensive ablation studies to analyze the various aspects of our proposed approach.

## 2 RELATED WORK

Unsupervised representation learning, with the promise of learning from large-scale unlabeled data, has drawn increasing attention in recent years, in both computer vision and natural language processing (NLP) communities. Most mainstream self-supervised methods can be categorized into three general directions: generative, denoising, and discriminative (Chen et al., 2020b; Grill et al., 2020; Doersch et al., 2015).

Generative and denoising methods seek to generate or reconstruct corrupted text/image/video tokens according to their empirical distributions. In generative and auto-regressive methods, next tokens are predicted given a causal context (Chen et al., 2020a; van den Oord et al., 2016) while denoising methods seek to reconstruct corrupted or masked tokens given an extended context (Devlin et al., 2019; Raffel et al., 2019). For text, since the tokens (words or sub-words (Sennrich et al., 2016; Wu et al., 2016)) are discrete and has relatively high entropy rate, language modeling has became the de-facto approach for pre-training models for most natural language tasks (Ruder et al., 2019). In the case of images, generative approaches often operate on pixel space (Bertalmio et al., 2001; Yu et al., 2018; Kim et al., 2019; Chen et al., 2020a; van den Oord et al., 2016), which can be extremely expensive for larger input size like videos and has hence limited the widespread adoption of these methods. Recently, discretizing images and videos with discrete variational auto-encoders (VQ-VAE), has been explored in compression and generative setups (van den Oord et al., 2017; Razavi et al., 2019; Walker et al., 2021; Ramesh et al., 2021; Yan et al., 2021), and Sun et al. (2019) tackles the video-language problem by frame-level quantization. Such approaches avoid modeling

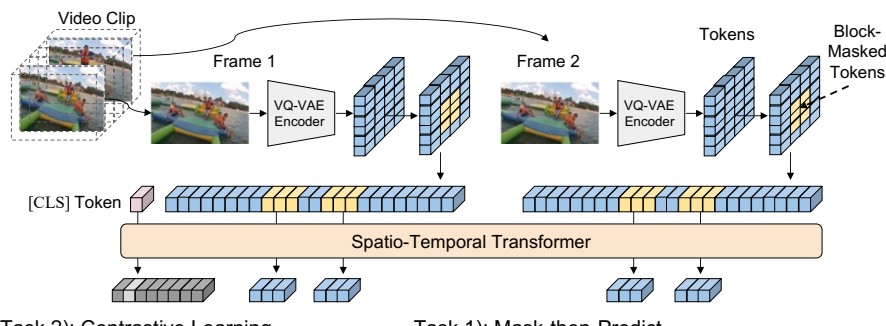

Figure 1: Overview of VIMPAC. Sampled video frames are discretized by VQ-VAE encoder into discrete tokens, which are then block-masked (in light yellow blocks). The model is self-supervised by two tasks: 1) *mask-then-predict* task predicts the masked tokens from their visible context; 2) *contrastive learning* task differentiates between positive and negative clips (details in Fig. 2) with [CLS] token feature. For brevity, we only show 2 frames with small token maps.

pixel-level details and have enabled the use of generative models for images and videos (Walker et al., 2021; Ramesh et al., 2021). Differing from these works, our framework investigates the use of such quantized representations in a denoising/reconstruction setup rather than generative, which has been shown in the NLP community to learn better representations (Raffel et al., 2019; Devlin et al., 2019). Moreover, beyond simply applying MLM to the video tokens, we propose a block masking strategy to reduce the strong local correlation in neighboring video tokens. This 3D block masking strategy is inspired from recent span-masking schemes (Raffel et al., 2019; Joshi et al., 2020) for language modeling. The concurrent work (Bao et al., 2021) explores using the VQ-VAE tokens as labels for masked patches in the image domain.

The other direction of research, which our framework combines, is discriminative methods which start from the hypothesis that learning to reconstruct local details is not necessary for learning good visual representations. In some of these approaches, an objective is constructed around hand-crafted heuristics tasks like spatial arrangement, color, playback speed or frame order predictions (Doersch et al., 2015; Zhang et al., 2016; Gidaris et al., 2018; Fernando et al., 2017; Lee et al., 2017; Wei et al., 2018; Epstein et al., 2020; Benaim et al., 2020; Sun et al., 2021). Another line of discriminative approaches is *contrastive learning* which aims at training a model to be able to recognize different views (e.g., different augmentation of images or different temporal samples of videos) of the same image or video, as a way to learn general representations (Chen et al., 2020b; He et al., 2020; Chen et al., 2020c; Grill et al., 2020; Caron et al., 2020; Feichtenhofer et al., 2021). This direction of research is reminiscent of sentence-order prediction tasks introduced in NLP (Devlin et al., 2019; Lan et al., 2019) with the goal of predicting whether two text sequences should be juxtaposed or not, an approach challenged in more recent literature (Liu et al., 2019; Lan et al., 2019). In the present work, inspired by visual representation rather than text representation learning literature, we adapt the contrastive learning approach to video by training a model to differentiate pairs of clips from a single video from pairs of clips from disparate videos.

Another thread of research focuses on the way to adapt transformer models to video tasks (Bertasius et al., 2021; Arnab et al., 2021). Recent works (Fan et al., 2021; Liu et al., 2021) extend it with hierarchical modeling and efficient attention patterns. These methods focus on the modeling and achieve good results with supervised pre-training. Besides the difference in focus (modeling *vs.* pre-training) to our paper, the hierarchical design is also not directly applicable to our mask prediction tasks.

## 3 METHODS

In this section, we present our proposed video pre-training method VIMPAC (Fig. 1) as well its detailed components. We first introduce the mask-then-predict task in Sec. 3.1, and then the contrastive learning task in Sec. 3.2. Lastly, we discuss how these two tasks are combined in Sec. 3.3.

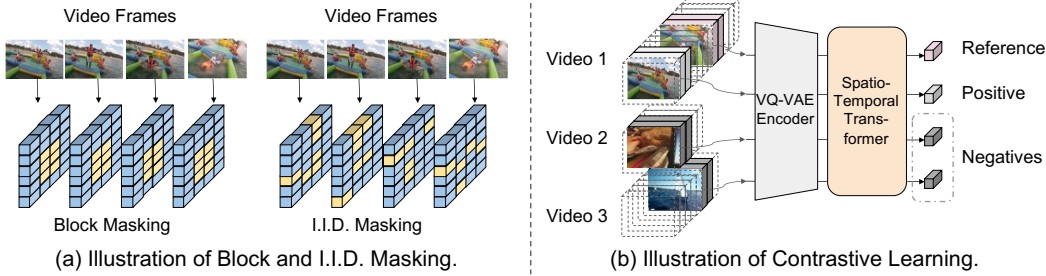

(a) Illustration of Block and I.I.D. Masking. | (b) Illustration of Contrastive Learning.

Figure 2: Illustration of pre-training tasks. *(a)*: block masking constructs the 3D-contiguous masking cube while i.i.d masking independently samples masked tokens. *(b)*: given the reference clip, the positive clip is uniformly sampled from the same video (video 1) while negative clips are sampled from other videos (video 2, 3). No spatial augmentations are applied to the raw video clips.

## 3.1 MASK-THEN-PREDICT TASK

Suppose that a video clip input comprises $T$ frames $\{f_1, f_2, \ldots, f_T\}$, the mask-then-predict task learns video representations by predicting the masked contents from their spatio-temporal context. Denote the set of mask-token locations as $M$, we learn to predict the original tokens $\{x_{t,i,j}\}$ (see details below) by optimizing the negative log-likelihood:

$$\mathcal{L}_{\text{mask}} = -\frac{1}{|M|} \sum_{t,i,j \in M} \log p_{t,i,j}\left(x_{t,i,j} \mid \{x_{t',i',j'}\}_{t',i',j' \in M^{\text{C}}}\right), \qquad (1)$$

where $M^{\text{C}}$ is the complement of $M$ and thus indicates the unmasked context.

**Video Quantization with VQ-VAE.** Since directly applying mask-then-predict over raw pixels and masking/predicting pixels leads to prohibitive computational costs and also tends to make the model overfit on detailed low-level visual information, we quantize the input videos with Vector Quantized-Variational Auto Encoder (VQ-VAE) (van den Oord et al., 2017; Ramesh et al., 2021). The VQ-VAE encoder takes an image as input and produces a token map, where the tokens belong to a predefined vocabulary $V$. The VQ-VAE decoder then tries to reconstruct the original image from these latent codes. In our method, we use a frozen and pretrained generic VQ-VAE encoder as a compressor that converts an input from an original input space $\mathbb{R}^{H \times W \times 3}$ into a discretized space $[V]^{\frac{H}{8} \times \frac{W}{8}}$. We independently apply the VQ-VAE encoder to each frame $f_t$ inside a clip. Specifically, we use the VQ-VAE trained in DALL-E (Ramesh et al., 2021). We keep the VQ-VAE weights frozen and do not finetune or adapt this model on our corpus.

**Block Masking** For sampling tokens to mask, the original BERT methods proposes the i.i.d. (independent and identically distributed) random mask $M_{\text{iid}}$ that constitutes of masked tokens:

$$M_{\text{iid}} = \{(t, i, j) \mid \mathcal{U}_{t,i,j}[0, 1] < \xi\}, \qquad (2)$$

where $\mathcal{U}_{t,i,j}[0, 1]$ is the uniform distribution from 0 to 1. Intuitively, $\xi$ is the expectation of masked-token ratio and hence controls the difficulty of our mask-then-predict task. In our early experiments, we found it easy to infer a masked token from its direct spatio-temporal neighbours (e.g., neighboring frames in a video tend to look similar thus contain similar tokens). To overcome this issue, we propose to use block masking (see Fig. 2 (a)), which masks continuous tokens inside spatio-temporal blocks. For each mask block $B$, we randomly sample lower ($B_{*,0}$) and upper boundaries ($B_{*,1}$) for each of the temporal ($T$), height ($H$), and width ($W$) dimensions. The direct product of the intervals delimited by these boundaries constructs the block mask. The final mask $M_{\text{block}}$ is the union of them:

$$M_{\text{block}} = \bigcup_B [B_{T,0}, B_{T,1}] \times [B_{H,0}, B_{H,1}] \times [B_{W,0}, B_{W,1}]. \qquad (3)$$

## 3.2 CONTRASTIVE LEARNING

Contrastive learning aims to distinguishing positive pairs from negative pairs (see Fig. 2 (b)). For each video $video_i$, we uniformly and independently sample two clips $c_i, c'_i$ as a positive pair, while

the clips in a batch belonging to other videos are used to construct negative pairs. A model (described in Sec. 3.4) processes clips $c_i$, $c'_i$ to build respective vector representations $f_i$, $f'_i$ and an InfoNCE (Oord et al., 2018) loss is used to distinguishes the positive feature pair $(f_i, f'_i)$ from the negative pairs $\bigcup \{\{(f_i, f_k), (f_i, f'_k)\} \mid k \neq i\}$ for each clip $c_i$:

$$\mathcal{L}_{\text{InfoNCE}}(i) = -\log \frac{\exp\left(f_i^\top f'_i / \gamma\right)}{\sum_{k \neq i} \exp\left(f_i^\top f_k / \gamma\right) + \sum_k \exp\left(f_i^\top f'_k / \gamma\right)}, \quad (4)$$

which we combine with the symmetric loss $\mathcal{L}'_{\text{InfoNCE}}(i)$ for paired clip sample $c'_i$. The final loss for a mini batch $\mathcal{L}_{\text{cl}}$ is the average loss for all $n$ clips in the mini-batch:

$$\mathcal{L}_{\text{cl}} = \frac{1}{n} \sum_{i=1}^{n} \mathcal{L}_{\text{InfoNCE}}(i) + \frac{1}{n} \sum_{i=1}^{n} \mathcal{L}'_{\text{InfoNCE}}. \quad (5)$$

### 3.3 Pre-Training Objective

We combine the two pre-training methods discussed above to define the overall objective as:

$$\mathcal{L} = \mathcal{L}_{\text{mask}} + \alpha \gamma \mathcal{L}_{\text{cl}}, \quad (6)$$

where $\alpha$ is a hyperparameter controlling the weight of the contrastive loss and multiplying the temperature $\gamma$ will smooth training (Grill et al., 2020; Chen et al., 2021). The inputs for both tasks are shared in mini-batches with the contrastive learning loss using the same block-masked inputs necessary for the mask-then-predict task. We highlight that the masked tokens are the only noise introduced in the contrastive learning, and that no other data augmentation is applied to raw pixels, in contrast to previous vision contrastive learning methods in which data-augmentation was paramount to the final performances of the model. This phenomenon is empirically studied in Sec. 5.2.3.

### 3.4 Modeling

The model architecture follows the standard transformer architecture in its post-layer-norm variant (Vaswani et al., 2017; Devlin et al., 2019) with two more recent additions: divided temporal-spatial attention (Bertasius et al., 2021), and sparse spatial attention (Child et al., 2019). We detail both additions in the Appendix. The model embedding layer maps the discrete tokens $\{x_{t,i,j}\}$ of a quantized input video (see Sec. 3.1) into dense vectors and sum them with positional embeddings. The backbone transformer model then outputs corresponding features $\{h_{t,i,j}\}$. We append an additional [CLS] token to each input sequence following Devlin et al. (2019) and use its output feature $h_{\text{cls}}$ as a representation for the whole video. For pre-training, we use two heads: a 2-layer MLP after each token outputs $\{h_{t,i,j}\}$ for the mask-then-predict task following BERT (Devlin et al., 2019), and a 3-layer MLP after the CLS output $h_{\text{cls}}$ for the contrastive learning task following SimCLR (Chen et al., 2020b). For fine-tuning on classification tasks, we remove the pre-training heads and add a fully-connected layer to the [CLS] output $h_{\text{cls}}$. More implementation details are in Appendix.

## 4 Experiments and Results

### 4.1 Datasets

For **pre-training**, we use the HowTo100M dataset (Miech et al., 2019) [1]. This dataset is constructed by searching YouTuBe videos with a list of text queries, it is significantly larger and more diverse than human-annotated datasets such as Kinetics 400 (Carreira et al., 2019). HowTo100M has 1.2M uncurated videos, with an average duration of 6.5 minutes. We only use videos and do not use other signals such as ASR captions in this dataset. For **downstream** evaluation, we experiment with several action classification datasets: UCF101 (Soomro et al., 2012a), HMDB51 (Kuehne et al., 2011a), Kinetics-400 (Carreira & Zisserman, 2017), SSV2 (Goyal et al., 2017), and Diving48 (Li et al., 2018). It is important to note that in many cases, actions in UCF101, HMDB51, and Kinetics-400 can be recognized from a single frame of the video, thus these datasets are '*spatially-heavy*'. As

---

[1] The VQ-VAE compression method is trained on 250M images as in (Ramesh et al., 2021). However, the VQ-VAE does not help with a better representation for the video pre-training as shown in Sec. 5.2.1, where directly training on the VQ-VAE tokens provides poor results.

Table 1: **Comparison with state-of-the-art.** Our model outperforms previous works on SSV2 and Diving48 dataset while showing competitive results on other datasets. Results on UCF101 and HMDB51 are average over three train-val splits. V,A,T refer to Visual, Audio, and Text modalities, respectively. [a](Grill et al., 2020; Feichtenhofer et al., 2021), [b](Miech et al., 2020), [c](Alayrac et al., 2020), [d](He et al., 2020; Feichtenhofer et al., 2021), [e](Bertasius et al., 2021), [f](Arnab et al., 2021), [g](Feichtenhofer et al., 2019), [h](Kalfaoglu et al., 2020), [j](Tran et al., 2018), [k](Wang et al., 2019), [l](Kondratyuk et al., 2021). K400=Kinetics-400 (Carreira & Zisserman, 2017), HT=HowTo100M (Miech et al., 2019), AudioSet (Gemmeke et al., 2017), IG-Uncurated (Ghadiyaram et al., 2019), IN21K=ImageNet-21K (Russakovsky et al., 2015). Note that some SotA models are pre-trained with extremely large (weakly-)supervised datasets, e.g., IG65M (Ghadiyaram et al., 2019) in [h]Kalfaoglu et al. (2020) and JFT-300M (Sun et al., 2017) in [f]Arnab et al. (2021).

| Method | Modality | Pre-Train Dataset | Temporally-Heavy | | Spatially-Heavy | | |
|---|---|---|---|---|---|---|---|
| | | | SSV2 | Diving48 | UCF101 | HMDB51 | K400 |
| *(Weakly) Supervised Pre-Training* | | | | | | | |
| K400 Sup. | V | K400 | $63.1^g$ | - | $96.8^j$ | $82.5^k$ | $81.5^l$ |
| TimeSformer[e] space-time | V | IN21K | 62.3 | 81.0 | - | - | 80.7 |
| TimeSformer[e] space-only | V | IN21K | 36.6 | - | - | - | 77.6 |
| ViViT[f] | V | IN21K/JFT300M | 65.4 | - | - | - | **84.8** |
| R(2+1)D BERT[h] | V | IG65M | - | - | **98.7** | **85.1** | - |
| *Self-supervised Pre-Training on Uncurated Videos* | | | | | | | |
| MIL-NCE[b] | V+T | HT | - | - | 91.3 | 61.0 | - |
| MMV[c] | V+A+T | AudioSet + HT | - | - | 95.2 | 75.0 | - |
| BYOL[a] | V | K400 | 55.8 | - | 96.3 | 75.0 | - |
| MoCo[d] | V | IG-Uncurated | 53.2 | - | 92.9 | - | - |
| **VIMPAC** | V | HT | **68.1** | **85.5** | 92.7 | 65.9 | 75.3 |

a consequence, image-level methods (Bertasius et al., 2021; Radford et al., 2021) show competitive results without modeling the temporal interactions inside the videos. To test the video model's ability beyond recognizing static images, we lay our focus on 'temporally-heavy' datasets (SSV2 and Diving48), in which action recognition from a single frame is more difficult. For example, it is almost impossible to distinguish two SSV2 classes *moving something up* and *moving something down* without reasoning across frames, and the same for different diving classes in Diving48. Additional dataset details (e.g., statistics) are presented in Appendix.

## 4.2 EXPERIMENTAL SETUP

Our model shapes follow BERT$_{LARGE}$ with 24 layers and hidden size 1024, but with halved attention head size and MLP intermediate size as in Child et al. (2019). For **pre-training**, we train the model for 100 epochs on HowTo100M with frames sampled at 2 FPS. We sample two clips from each video as model inputs as described in Sec. 3.2. To reduce computation cost, we train the first 90 epochs with a smaller input resolution (#frames $T$=5 and frame size $S$=128) and increase the spatial resolution ($T$=5, $S$=256) for the last 10 epochs following Devlin et al. (2019). Positional embeddings are interpolated as in Dosovitskiy et al. (2021) when input resolution changes. Importantly, our pre-training scheme does not involve spatial augmentations: all frames are resized and centered cropped without random flipping, color distortion, etc. We use a batch size of 1024 in pre-training. The number of negative clips used for contrastive learning is 255 for the first 90 epochs and 127 for the last 10 epochs. The number of negative pairs used in our ablation analyses is kept constant at 127. More details are in Appendix. For **fine-tuning**, we use more input frames ($T$=10 and $S$=256), and batch size 128. We sample frames at 2 FPS for datasets with longer videos (i.e., UCF101 and Kinetics-400), and sample 4 FPS for datasets with shorter videos (i.e., HMDB51, SSV2, Diving48). During inference, we follow Feichtenhofer et al. (2019; 2021) to use 3 spatial crops and 10 temporal crops (in total 30 crops), and average their prediction scores as the final score.[2] All models are trained with AdamW (Loshchilov & Hutter, 2018) optimizer with linear warm-up and linear learning rate decay. We observe similar pre-training instability as reported in Chen et al. (2020a; 2021) and follow their practice to sequentially choose learning rate at 1e-3, 5e-4, 3e-4, ..., until convergence.

---

[2]As in Bertasius et al. (2021); Arnab et al. (2021), we observe that the performance is saturated at 4~5 temporal crops for our model.

Table 2: Impact of **model size**. 'Speed' is the normalized pre-training speed measured by #videos/second on one V100 GPU. 'Mask-Accu.' and 'CL-Loss' are mask-then-predict accuracy and contrastive learning loss to indicate the pre-training performance. 'UCF101' is the fine-tuning accuracy on UCF101 dataset. By default, we use the configuration in the first line in our analysis. The configuration that produced the final results are underlined.

| Layers | Dim | Params | Speed | Mask-Accu.↑ | CL-Loss ↓ | UCF101↑ |
|--------|-----|--------|-------|-------------|-----------|---------|
| 6 | 512 | 29.4M | 32.0 | 17.2 | 1.06 | 69.4 |
| 6 | 768 | 63.0M | 21.0 | 17.7 | 1.03 | 75.0 |
| 12 | 512 | 54.7M | 18.1 | 17.9 | 1.02 | 76.6 |
| 12 | 768 | 119.7M | 11.2 | 18.4 | 1.00 | 78.1 |
| 24 | 1024 | 210.1M | 5.0 | 18.7 | 0.98 | 78.5 |

## 4.3 RESULTS

Table 1 shows our primary results. We mainly compare with self-supervised pre-training methods on uncurated videos. In addition, we list methods with (weakly) supervised pre-training as references, though they can not be fairly compared with our method as they use large-scale (weakly) labeled data. As discussed in Sec. 4.1, recognizing actions in SSV2 and Diving48 require a strong temporal reasoning ability, while in the other datasets, spatial understanding is dominant. To better illustrate the differences between temporally-heavy and spatially-heavy datasets, we compare two variants of TimeSformer (Bertasius et al., 2021), one with attention on space-time, and one on space only. Note the gaps between these two variants are significantly larger for temporally-heavy datasets (SSV2) than spatially-heavy datasets (Kinetics-400), demonstrating the importance of temporal modeling for temporally-heavy datasets.[3] On the two temporally-heavy datasets SSV2 and Diving48, when comparing to previous best models among all self-supervised and supervised pre-training methods, our model VIMPAC sets new state of the art, where we achieve 2.7% and 4.5% absolute improvement, respectively. This is especially surprising considering the two previous SotA models ViViT (Arnab et al., 2021) and TimeSformer (Bertasius et al., 2021) both use large-scale supervised pre-training, and ViViT also uses various regularization techniques (e.g., stochastic depth (Huang et al., 2016), random augment (Cubuk et al., 2020), and mixup (Zhang et al., 2018)). On spatially-heavy datasets,UCF101, HMDB51 and Kinetics-400, VIMPAC achieves competitive results to self-supervised pre-training methods, while being lower when compared to supervised methods. These relatively low results of our VIMPAC (e.g., UCF101) are possibly due to the spatial information loss during the VQ-VQA quantization process. Concurrent work BEiT (Bao et al., 2021) addresses this issue by using image patches instead of VQ-VAE tokens as inputs, where they show strong performance on image classification tasks. We encourage future work to study using image patches as inputs for better spatial modeling under our framework. Previous self-supervised pre-training methods such as BYOL (Grill et al., 2020; Feichtenhofer et al., 2021) and MoCo (He et al., 2020; Feichtenhofer et al., 2021) are good at global understanding, but the pre-training schema does not consider the internal interactions inside videos (especially for the temporal dimension). As a result, it could reach or even outperform the supervised alternatives on UCF101. However, it shows lower results on SSV2 compared to the transformers (Bertasius et al., 2021; Arnab et al., 2021) (although with different backbones) that warm up from image-pre-trained models and learn the temporal interactions directly from the downstream tasks. We also show the cross-modal self-supervised learning methods, MIL-NCE (Miech et al., 2020) and MMV (Alayrac et al., 2020) that are trained on uncurated videos but leverage other modalities (e.g., text) to help video learning.

## 5 ANALYSIS

We also analyze the model's scalability and the effectiveness of our pre-training methods. To save computation, for all analyses, we use a smaller model (6-layer transformer with hidden dimension 512) and smaller input resolution (5 input frames with spatial size 128, i.e., $T$=5, $S$=128) throughout this section, unless otherwise stated. We also perform pre-training with fewer epochs (i.e., 10). For downstream tasks, we use the same input resolution as pre-training (i.e., $T$=5, $S$=128), and we use 2 temporal crops for inference. All results are reported on the train-val split 1 if applicable.

---

[3]The image model CLIP (Radford et al., 2021) achieves 92.0% on the spatially-heavy UCF-101 dataset.

Table 4: Impact of **masking strategy**. Models are pre-trained with only mask-then-predict.

| Strategy | Frame Size $S$ | Mask-Accu.↑ | UCF101 ↑ |
|---|---|---|---|
| block | 128 | 17.6 | 68.3 |
| i.i.d. | 128 | 24.3 | 63.5 (-4.8) |
| block | 256 | 11.2 | 69.5 |
| i.i.d. | 256 | 19.5 | 61.4 (-8.1) |

Table 5: Impact of **masking ratio**. Models are pre-trained with only mask-then-predict. Default setup is underlined.

| Strategy | #Blocks | Ratio | Mask-Accu.↑ | UCF101 ↑ |
|---|---|---|---|---|
| block | 4 | 11.9% | 17.9 | 66.8 |
| block | 5 | 14.5% | 17.6 | 68.3 |
| block | 6 | 17.0% | 17.3 | 67.3 |

## 5.1 SCALABILITY

In Table 2, we illustrate the scalability of our method with different model sizes (i.e., number of layers and hidden dimensions). Larger models have more parameters ('Params') and higher computational cost (measured by the normalized pre-training 'Speed'). To evaluate the pre-training tasks performance, we provide both pre-training metrics (mask-then-predict accuracy denoted by 'Mask-Accu.', and contrastive learning loss denoted by 'CL-Loss') and UCF101 downstream fine-tuning results. As the size of the model grows, the fine-tuning results show consistent improvement with the pre-training metrics. Note that for the last row in Table 2, we halve the attention head and MLP intermediate dimensions. We also illustrate the scalability over input resolution in Appendix F.2.

## 5.2 PRE-TRAINING METHODS

### 5.2.1 THE IMPACT OF PRE-TRAINING

We first compare different pre-training tasks and the non-pre-training results. As shown in Table 3, mask-then-predict is good at temporally-heavy datasets (SSV2, Diving48) while contrastive learning improves the spatially-heavy datasets.[4] We also compare with the non-pre-training results (the first row of Table 3) and observe that both tasks significantly improve the results. We notice that these non-pre-training results are lower than previous from-scratch models, which might be caused by the difficulty in

Table 3: Impact of **pre-training tasks**. 'MP'=Mask-then-Predict, 'CL'=Contrastive Learning task.

| MP | CL | Temporally-Heavy | | Spatially-Heavy | | |
|---|---|---|---|---|---|---|
| | | SSV2 | Diving48 | UCF101 | HMDB51 | K400 |
| ✗ | ✗ | 1.2 | 10.0 | 41.3 | 19.0 | 41.0 |
| ✗ | ✓ | 32.5 | 26.3 | 57.1 | 30.7 | 47.0 |
| ✓ | ✗ | 41.4 | 37.2 | 68.3 | 35.3 | 53.7 |
| ✓ | ✓ | 41.1 | 37.5 | 69.4 | 37.8 | 54.5 |

training video transformers (Bertasius et al., 2021; Arnab et al., 2021) and the information loss in our input quantization process (Ramesh et al., 2021).

### 5.2.2 MASK-THEN-PREDICT

In this analysis, we exclude the contrastive learning loss (i.e., loss weight $\alpha$=0) to avoid side effects.

**Block Masking versus I.I.D. Masking.** We first compare our proposed block masking strategy and the uniform i.i.d. masking strategy (discussed in Sec. 3.1 and illustrated in Fig. 2). As shown in Table 4, although the i.i.d. masking achieves higher pre-training mask-token-prediction accuracy ('Mask-Accu.'), it shows lower downstream results ('UCF101') than block masking. The higher mask accuracy is possibly due to the easier i.i.d. mask-then-predict task. We show in Appendix that simply copy-paste already yield reasonable reconstruction results. The existence of such a trivial solution potentially prevents the model from learning useful video representations for downstream tasks. Meanwhile, we also find that the model with larger input frame size 256 benefits more from the block masking strategy, because the adjacent tokens are closer in the original 2D image for these larger frames. Hence, the spatial locality is amplified.

**Masking Ratio.** In Table 5, we study the impact of masking ratio, by varying the number of masked blocks for block masking. Empirically, the result differences among different masking ratios are marginal and the original BERT's 15% masking ratio (with roughly 5 masking blocks) works slightly

---

[4]Empirically, we observe that the improvement of contrastive learning (CL) becomes higher when training with more epochs and larger architectures (as also shown in Chen et al. (2020b); Qian et al. (2021)). With the Base model (but also using decreased training epochs), CL gives a 3% improvement on UCF101 over the pure mask-then-predict pre-training. In this section, we provide the comprehensive ablation studies based on the small model because of the limited budget.

Table 6: Impact of **maximum sampling distance** $d_{max}$ (sec.) between two positive clips.

| $d_{max}$ | Mask-Accu.↑ | CL-Loss↓ | UCF101↑ |
|---|---|---|---|
| $\infty$ | 17.2 | 1.06 | 69.4 |
| 30 | 17.3 | 0.77 | 69.0 (-0.4) |
| 10 | 17.4 | 0.61 | 68.3 (-1.1) |
| 0 | 17.5 | 0.41 | 66.7 (-2.7) |

Table 7: Impact of **#negative samples**.

| #samples | Mask-Accu.↑ | CL-Loss↓ | UCF101↑ |
|---|---|---|---|
| 128 -1 | 17.2 | 1.06 | 69.4 |
| 256 - 1 | 17.1 | 1.30 | 69.2 |
| 512 - 1 | 17.2 | 1.56 | 70.4 |
| 1024 - 1 | 17.0 | 1.86 | 69.8 |

better. Thus we always select the number of mask blocks whose induced masking ratio is closest to 15%. The detailed choices of number of masking blocks are listed in Appendix.

### 5.2.3 CONTRASTIVE LEARNING

**Positive Sampling Distance.** As illustrated in Sec. 3.2 and Fig. 2.(b), we uniformly sample positive clip pairs across the whole video without any distance restriction. To analyze the effect of such a sampling strategy, We perform a set of experiments by varying the maximum sampling distance $d_{max}$ (in seconds) between two positive clips. The results are shown in Table 6. $d_{max}=\infty$ denotes our default setup without any distance restriction. $d_{max}=0$ samples two same clips, and $d_{max}=10$ samples two positive clips with a maximum distance of 10 seconds. Although previous contrastive learning methods (Qian et al., 2021; Feichtenhofer et al., 2021) favor the sampling of temporal positives within a shorter range (e.g., maximum 36 seconds for uncurated videos in Feichtenhofer et al. (2021)), we observe a performance gain when using larger distance. We also want to emphasize that the results with $d_{max}=10$ and $d_{max}=0$ are not better than the model pre-trained with only mask-then-predict (UCF101 accuracy 68.3), which suggests that short-range contrastive learning does not improve upon our mask-then-predict task. This is potentially because our mask-then-predict already gives the model the ability to model local interactions, thus contrastive learning objective can only be useful when it focuses on longer-range interactions.

**Number of Negative Samples.** Previous constrastive learning methods (Chen et al., 2020b; 2021; Feichtenhofer et al., 2021) benefit from more negative samples. In this section, we show that the number of negative samples has less impact on our method when mask-then-predict task is added. As shown in Table 7, we use different contrastive learning sample sizes (i.e., $n$ in Sec. 3.2) and always accumulate the gradients to 1024 samples before updating the parameters. Although increasing sample size makes the contrastive learning task harder (reflected by 'CL-Loss'), it does not show clear evidence of improving UCF101 downstream performance.

**Input Masking as Augmentation.** Most self-supervised visual representation learning methods (Chen et al., 2020c;b; Grill et al., 2020; Feichtenhofer et al., 2021; Qian et al., 2021) based on contrastive learning suffer from a large drop when removing strong spatial augmentations. In contrast, our pre-training does not use any spatial augmentations on raw frames. However, as we tie the input between mask-then-predict and contrastive learning to reduce computation cost, the random masking noise is naturally introduced. We here investigate its impact in Table 8.

Table 8: Impact of **mask augmentation in contrastive learning**. 'MP'=Mask-then-Predict. 'CL-Mask'=Use input mask in CL. Default setup is underlined.

| MP | CL-Mask | Mask-Accu.↑ | CL-Loss↓ | UCF101↑ |
|---|---|---|---|---|
| ✗ | ✗ | - | 1.07 | 57.1 |
| ✗ | ✓ | - | 1.08 | 55.5 |
| ✓ | ✗ | 17.2 | 1.04 | 67.4 |
| ✓ | ✓ | 17.2 | 1.06 | 69.4 |

When pre-trained jointly with mask-then-predict, adding mask noise improves UCF101 accuracy by +2.0; however, when pre-trained without it, adding mask noise hurts the performance (-1.6). We hypothesize that this is due to the large input mismatches between pre-training and fine-tuning when mask-then-predict objective is not applied. Noisy masking creates 'holes' to the input token maps during pre-training, while for fine-tuning the input token maps are intact.

## 6 CONCLUSION

We present the video pre-training framework VIMPAC that introduces mask-then-predict task to video self-supervised learning. mask-then-predict task helps model spatio-temporal interactions that is important for video understanding. We use the VQ-VAE quantizer and propose the block mask-

ing method that is essential to overcome the strong locality in video. The contrastive learning task is also added to learn separable global features. Different from previous methods, our contrastive learning does not use data augmentation over raw frames and is less sensitive to the temporal sampling distribution for positive pairs. We show that our frameworks could achieve state-of-the-art performance on two temporally-heavy dataset (SSV2 and Diving48) and reach competitive results on other datasets. Detailed analysis is provided regarding the model scalability and task design.

**Reproducibility statement.** The code to reproduce the results in this paper is submitted in the supplementary material and will be made public. The code is runnable and contains the scripts with faithful hyperparameters for our experiments. We include comprehensive instructions about the feature extraction, pre-training, and fine-tuning. A detailed "readme" file is attached in the codebase, and we will publicly release the pre-trained weights as well.

**Ethical considerations and limitations.** The main purpose of this work is to design a video pre-training method to capture internal interactions. In real life, a lot of tasks are temporally-heavy such that the task completion relies on understanding the past. Video understanding is a proxy to these embodied studies eventually and thus could one day benefit our daily life. Our work is also potentially useful for recovering corrupted videos (but this could also be possibly misused, hence we recommend careful and safe use of this technology, including previous works). In this paper, we mainly consider the action classification tasks with different characteristics (i.e., spatially-heavy and temporally-heavy) and we show the recovering ability of our model. It's also possible to employ VIMPAC backbone to video detection tasks where tokens' outputs can be viewed as anchors. The DALL-E VQ-VAE code and model are published under MIT license. We are among the first line of work that use masked token reconstruction, etc. as a self-supervised learning approach for vision, and we do agree the the use of VQ-VAE tokens might limit the models' ability in certain aspects, such as spatial info loss. Future work could explore combining the techniques from BEiT (Bao et al., 2021) for a more capable approach. Meanwhile, token-level approach has its unique advantage of generative modeling (as shown in VQ-GAN (Esser et al., 2021), DALL-E (Ramesh et al., 2021)), more robust to the input noise, and has the possibility to transfer to other modality (e.g., text; since they share the same types of input). This cross-modality transferability is another future direction that we consider.

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

In this supplementary materials, we start with describing details of the model (Sec. A), pre-training (Sec. B), experiments (Sec. C), and dataset (Sec. D). We then provide additional analysis results in Sec. F and visualization in Sec. G.

# A    MODEL ARCHITECTURE

As described in Sec. 3.4, we use a transformer model on top of the discrete video tokens generated by VQ-VAE. Since transformers have different variants, we here show details of our architecture for clarity. The design of our method largely follows the practice in BERT (Devlin et al., 2019), TimeSformer (Bertasius et al., 2021), Sparser Transformer (Child et al., 2019), ViViT (Arnab et al., 2021), and MoCoV3 (Chen et al., 2021).

## A.1    BACKBONE MODEL

**Embedding.**    Given the video frames $\{f_t \in \mathbb{R}^{H \times W} \mid t \in [T]\}$, we first use VQ-VAE (van den Oord et al., 2017; Ramesh et al., 2021) to discrete them into video tokens $\{x_{t,i,j} \in [V] \mid t \in [\hat{T}], i \in [\hat{H}], j \in [\hat{W}]\}$. We use the specific VQ-VAE in DALL-E (Ramesh et al., 2021) which is trained on trained on 250 million images from the Internet. Since the VQ-VAE encoder largely compresses a 8x8x3 vector (ranging from 0-255) to an integer of 0-8191, it is considered as a compression method with image prior. We do not use the Video-VQVAE (Walker et al., 2021) method since the image-trained VQVAE has been pretrained on a very large image corpus and as a consequence cover a much more diverse set of visual scenes and elements. We next use an embedding layer (*embedding*) that to map these discrete tokens to continuous vectors. Since transformer layers are permutation-invariant, we follow (Dosovitskiy et al., 2021; Devlin et al., 2019) to add positional information into the input. The positional embedding (*pos*) is factorized as a sum of the temporal embedding $pos^{\mathrm{T}}$, the height embedding $pos^{\mathrm{H}}$, and the width embedding $pos^{\mathrm{W}}$. This factorization reduces the number of trainable parameters to encode positional information, which empirically shows a slightly better result. Finally, a Layer-Normalization (Ba et al., 2016) layer is added to get the initial hidden outputs $h_{t,i,j}^0$:

$$h_{t,i,j}^0 = \mathrm{LayerNorm}(embedding(x_{t,i,j}) + pos(t,i,j)), \tag{7}$$

$$pos(t,i,j) = pos^{\mathrm{T}}(t) + pos^{\mathrm{H}}(i) + pos^{\mathrm{W}}(j), \tag{8}$$

where we use the superscript 0 to denote that it is the hidden outputs before the first transformer layer.

**Attention Blocks.**    Before introducing the detailed model architecture, we first describe the basic building components: the attention block. An attention block is built based on the attention operator (i.e., 'Attn') with a residual connection. The attention operator takes a single query vector $x$ and its context $\{y_i\}$ as input. It first computes the attention score between $x$ and each context vector $y_i$, then the attention scores are normalized by the $\mathrm{softmax}$. Lastly, the output is a weighted-sum over all the context vectors (transferred by a 'value' matrix $W_{\text{value}}$):

$$\mathrm{Attn}(x, \{y_i\}) = \sum_i \mathrm{softmax}_i\{(W_{\text{query}}\, x)^\top W_{\text{key}}\, y_i\} W_{\text{value}}\, y_i. \tag{9}$$

To compose the attention block from the previous attention operator, the residual connection and layer normalization (i.e., 'LayerNorm') are added. We follow the original transformer model (Vaswani et al., 2017) that uses a post-layer-norm layout:

$$\mathrm{AttnBlock}(x, \{y_i\}) = \mathrm{LayerNorm}(x + W_{\text{out}}\, \mathrm{Attn}(x, \{y_i\})). \tag{10}$$

In order to reduce computational cost and memory, we also adapt the attention block suggested in Sparse Transformer (Child et al., 2019) that takes two sets of context vectors $\{y_i\}$ and $\{z_j\}$ as input. This special attention block computes attention for the two context-vector sets separately and concatenates their output together. In our case, suppose $\{y_i\}$ and $\{z_j\}$ are the rows and columns of a square matrix, then it reduces the computation cost of calculating attention scores from $\Theta(n^4)$ to $\Theta(n^2)$, where $n$ is the number of rows/columns:

$$\text{AttnBlock}(x, \{y_i\}, \{z_j\}) = \text{LayerNorm}(x + W_{\text{out}}[\text{Attn}(x, \{y_i\}), \text{Attn}(x, \{z_j\})]) \tag{11}$$

**Spatio-Temporal Transformer Layer.** The spatio-temporal transformer layer is composed with the previously-introduced attention blocks and an additional MLP block. The $l$-th layer takes the output of the previous layer $\{h_{t,i,j}^{l-1}\}$ as input and outputs the hidden states $\{h_{t,i,j}^l\}$. We separate the attention into two attention blocks: the temporal attention block $\text{AttnBlock}_{\text{TIME}}$ and the spatial attention block $\text{AttnBlock}_{\text{SPACE}}$. Without loss of generality, we will use $g^{\text{TIME}}$ and $g^{\text{SPACE}}$ to denote the intermediate results from temporal and spatial attention blocks, respectively. First, the temporal attention block attends to the tokens at the same spatial location but in different frames (i.e., at different timesteps): $\{h_{t,i,j}^{l-1} \mid t \in [T]\}$. Next, the spatial attention block attends to the tokens in the same frame: $\{g_{t,i,j}^{\text{T}} \mid (i,j) \in [\hat{H}] \times [\hat{W}]\}$. To reduce the computational cost, we incorporate the sparse attention block (Child et al., 2019) (detailed in the previous paragraph) that factorizes the attention over height and width: $\{g_{t,i,j}^{\text{T}} \mid i \in [\hat{H}]\}, \{g_{t,i,j}^{\text{T}} \mid j \in [\hat{W}]\}$. The MLP block has two fully-connected layers with GeLU (Hendrycks & Gimpel, 2016) activation in the middle. Overall, the formula of one spatio-temporal transformer layer is:

$$g_{t,i,j}^{\text{TIME}} = \text{AttnBlock}_{\text{TIME}}(h_{t,i,j}^{l-1}, \{h_{t,i,j}^{l-1} \mid t \in [T]\}) \tag{12}$$

$$g_{t,i,j}^{\text{SPACE}} = \text{AttnBlock}_{\text{SPACE}}(g_{t,i,j}^{\text{T}}, \{g_{t,i,j}^{\text{TIME}} \mid i \in [\hat{H}]\}, \{g_{t,i,j}^{\text{TIME}} \mid j \in [\hat{W}]\}) \tag{13}$$

$$h_{t,i,j}^l = \text{LayerNorm}(g_{t,i,j}^{\text{SPACE}} + \text{MLP}(g_{t,i,j}^{\text{S}})) \tag{14}$$

**[CLS] Token.** Following the practice in BERT (Devlin et al., 2019) design, we add a special [CLS] (abbreviation of 'classification') token and take its output as the representation of the whole sequence. We follow TimeSformer (Bertasius et al., 2021) to compute the its output: the [CLS] token attends over the context separately and then the outputs are averaged. We take the temporal attention layer as an example. Suppose $h_{\text{cls}}^{l-1}$ is the [CLS] feature vector output by layer $l-1$, then the temporal attention layer do the following computation:

$$g_{\text{cls}}^{\text{TIME}} = \frac{1}{\hat{H}} \frac{1}{\hat{W}} \sum_i \sum_j \text{AttnBlock}_{\text{TIME}}(h_{\text{cls}}^{l-1}, \{h_{t,i,j}^{l-1} \mid t \in [T]\}). \tag{15}$$

The other attention blocks process the [CLS] token similarly.

## A.2 Pre-Training and Fine-Tuning Heads

Pre-training or fine-tuning usually requires a few additional modules (i.e., heads) on top of the transformer layers that convert the output features to the desired probabilities or vectors. We next describe the heads used in our pre-training and fine-tuning process.

**Token Head for Mask-then-Predict.** We first define the prediction head over the tokens following BERT(Devlin et al., 2019). It first processes the last-layer hidden outputs $h_{t,i,j}^L$ using a fully-connected layer (with GELU activation (Hendrycks & Gimpel, 2016) and layer normalization (Ba et al., 2016)):

$$u_{t,i,j} = \text{LayerNorm}\left(\text{GELU}(W_{\text{token}}(h_{t,i,j}^L) + b_{\text{token}})\right). \tag{16}$$

In our mask-then-predict method (Sec. 3.1), we will predict the masked tokens (i.e., the token before masking) from their context. We thus further convert this hidden vector into a distribution over the token vocabulary:

$$P_{t,i,j}(o_{t,i,j} = k) = \text{softmax}_k\{W_{\text{word}} u_{t,i,j} + b_{\text{word}}\}. \tag{17}$$

The weight $W_{\text{word}}$ is shared with input word embedding layer *embedding* (Press & Wolf, 2017; Devlin et al., 2019) while the bias $b_{\text{word}}$ is trained independently.

Table 9: **Induced Masking ratio** w.r.t. to different input resolutions and #masking blocks. The numbers of blocks/masking ratio for each resolution setting used in our experiments are shown in **bold**.

| | Input Resolution | | #Masking Blocks | | | | |
|---|---|---|---|---|---|---|---|
| Length | Frame Size | Token Map Size | 4 | 5 | 6 | 7 | 8 |
| 5 | 128 | 16 | 11.9 | **14.5** | 17.0 | 19.4 | 21.7 |
| 5 | 256 | 32 | 10.6 | 13.1 | **15.2** | 17.5 | 19.5 |
| 10 | 128 | 16 | 10.4 | 12.8 | **15.0** | 17.1 | 19.2 |
| 10 | 256 | 32 | 9.3 | 11.4 | 13.4 | **15.4** | 17.2 |

**Contrastive Learning Head** Next we discuss the pre-training heads for contrastive learning. It is on top of the `[CLS]` hidden output $h_{\text{CLS}}$. We encode the hidden state with MLP. We use batch normalization (Ioffe & Szegedy, 2015) inside the MLP head following the practice in (Chen et al., 2021).

$$f_{\text{CLS}} = \text{MLP}_{\text{CLS}}(h_{\text{CLS}}) \tag{18}$$

This $f_{\text{CLS}}$ feature is used in computing the contrastive loss as in Sec. 3.2.

**FC Layer for Fine-Tuning.** When fine-tuning for action classification task, we add a fully-connected (FC) layer to the `[CLS]` output $h_{\text{cls}}$. We initialize its weight and bias to zero.

**Special Tokens.** Besides the $V$ token types introduced in the vocabulary of the VQ-VAE (see Sec. 3.1), we add several special tokens into the 'vocabulary', namely a `[CLS]` token is introduced as a stub for the whole-video representation, a `[PAD]` token is used when the actual clip length is less than the model's expected input length. For the mask-then-predict task, we follow BERT (Devlin et al., 2019) to replace the masked tokens with a specific `[MASK]` token.

# B PRE-TRAINING DETAILS

## B.1 MASKING BLOCKS

As described in Sec. 3.1, we mask the tokens by blocks (a cube-shape set of tokens). To avoid masking all the tokens in the clip, we control the maximum block length for the time domain, height, and width. For spatial dimensions (i.e., height and width), the maximum length is half of the full length (e.g., the maximum block length will be 16 for a token map of length 32). For temporal dimension (i.e., the clip length), the maximum length will be 2/3 (round up) of the full length so that it allows long-range modeling. Under these constraints, we uniformly sample a fixed number of mask blocks and take their union to construct the final mask. The number of blocks is decided by the induced masking ratio, which depends on the input resolutions. In Table 9, we show the induced masking ratio w.r.t. different input resolutions and #masking blocks. We take the VQ-VAE (van den Oord et al., 2017) provided in DALL-E (Ramesh et al., 2021) that has a compression factor of 8, thus the length of the token map is always 1/8 of the frame size. For each input resolution, we select the number of blocks (shown in bold in Table 9) whose induced masking ratio is closet to 15% following BERT (Devlin et al., 2019).

## B.2 CONTRASTIVE LEARNING LOSS

For completeness, we list the two losses used in contrastive learning here. The first loss for clip $c_i$ from video$_i$ is:

$$\mathcal{L}_{\text{InfoNCE}}(i) = -\log \frac{\exp\left(f_i^\top f_i'/\gamma\right)}{\sum_{k \neq i} \exp\left(f_i^\top f_k/\gamma\right) + \sum_k \exp\left(f_i^\top f_k'/\gamma\right)} \tag{19}$$

The symmetric loss $\mathcal{L}'_{\text{InfoNCE}}(i)$ for feature of the other clip sample $c_i'$ from video$_i$ (and its feature $f_i'$) is:

$$\mathcal{L}'_{\text{InfoNCE}}(i) = -\log \frac{\exp\left(f_i'^\top f_i/\gamma\right)}{\sum_k \exp\left(f_i'^\top f_k/\gamma\right) + \sum_{k \neq i} \exp\left(f_i'^\top f_k'/\gamma\right)} \tag{20}$$

Table 10: **Model Configuration**. The 'Small' model is mainly used in the analysis (Sec. 5) while 'Large-Half' model is mainly used in the results (Sec. 4.3) for the final large-scale experiments. 'Vocab Size' is the number of token types in our model, defined by the pre-trained VQ-VAE model (Ramesh et al., 2021).

|  | Small (in Sec. 5) | Base | Large-Half (in Sec. 4.3) |
|---|---|---|---|
| Layers | 6 | 12 | 24 |
| Dimensions | 512 | 768 | 1024 |
| Attention Heads | 8 | 12 | 16 |
| Attention Head Dim | 64 | 64 | 32 |
| MLP Intermediate Size | 2048 | 3072 | 2048 |
| Vocab Size | 8192 | 8192 | 8192 |
| Params | 29.4M | 119.7M | 210.1M |

Table 11: **Training Hyperparameters**. 'Pre-Train (Results)' is our final model in Sec. 4.3 that takes a large-half model. 'Pre-Train (Analysis)' is the pre-training in analysis (Sec. 5). *The batch size for pre-training is the number of samples in updating the weights, Since we use gradient accumulation, it is not correlated to the number of negative examples in contrastive learning.

|  | Pre-Train (Results) | Pre-Train (Analysis) | SSV2 | Diving48 | UCF101 | HMDB51 | Kinetics-400 |
|---|---|---|---|---|---|---|---|
| *Optimization* | | | | | | | |
| Number of Epochs | 100 | 10 | 22 | 50 | 50 | 50 | 30 |
| Number of Updates | 120K | 12K | 29K | 5.8K | 3.7K | 1.4K | 48K |
| Learning Rate | 3e-4 | 1e-3 | 1e-4 for small/base model, 5e-5 for large-half model | | | | |
| Warm-Up Ratio | 0.05 | 0.1 | 0.1 | | | | |
| LR Decay | Linear | | Linear | | | | |
| Backbone Dropout | 0.1 | | 0.1 | | | | |
| Last FC Dropout | - | | 0.0 | | | | |
| Optimizer | AdamW | | AdamW | | | | |
| Batch Size | 1024* | | 128 | | | | |
| Weight-Decay | 0.05 | | 0.01 | | | | |
| Adam Beta1 | 0.9 | | 0.9 | | | | |
| Adam Beta2 | 0.98 | | 0.999 | | | | |
| Adam Epsilon | 1e-8 | | 1e-8 | | | | |
| Grad-Clipping Norm | 1.0 | | 1.0 | | | | |
| *Data Augmentation* | | | | | | | |
| Color Distortion/Gray-Scale | No | | No | | | | |
| Training Spatial Resize | 1 (Frame Size) | | 2 (Frame Size, Frame Size * 1.25) | | | | |
| Training Spatial Crops | 1 (Center) | | 3 (Top-Left, Center, Bottom-Right) | | | | |
| Training Temporal Crops | 2 (Random Uniform) | | 1 (Random Uniform) | | | | |
| Inference Spatial Resize | 1 (Frame Size) | | 1 (Frame Size) | | | | |
| Inference Temporal Crops | 1 (Random Uniform) | | 10 (Uniform) | | | | |
| Training Spatial Flip | No | | No | Yes | | | |
| Inference Spatial Crops | 1 (Center) | | 1 (Center) | 3 (Top-Left, Center, Bottom-Right) | | | |

## C  EXPERIMENT DETAILS

In this section, we show our model configuration and training hyperparameters in details to support the reproducibility of our experiments.

**Model Configuration.**    Our model configuration details is shown in Table 10. Most analysis results (Sec. 5) take 'Small' models and our final results (Sec. 4.3) take 'Large-Half' model. Other models are used in Sec. 5.1. The final 'Large-Half' model halves the attention head dimension and MLP intermediate size as in (Child et al., 2019). For the pre-training heads, we follow BERT to take the intermediate dimension of the token-head to be the same as the backbone's hidden dimension. For the CLS head, we take 3 layers in MLP and 4096 intermediate dimensions. The output dimension is 256. We test with different number of layers and hidden dimensions of CLS head and generally find that larger head gives better results (as in (Qian et al., 2021; Chen et al., 2021)). This CLS head contributes to about $1\%$ pre-training computational cost overhead.

**Training Hyperparameters.**    We list the training hyperparameters in Table 11. Most of the hyperparameters are inherited from previous works to allow fair comparison and reduce tuning effort. For optimizer hyperparameters, we mostly follow the implementation of DALL-E (Ramesh et al., 2021) and BERT (Devlin et al., 2019). SSV2 follows the epoch number in (Feichtenhofer et al., 2019) and (Feichtenhofer et al., 2021). To reduce the computational cost, we pre-extract the VQ-VAE tokens thus we employ a fixed set of spatial data augmentations. As listed in the bottom of Table 11, we exclude any color distortion and gray scale augmentation. We resize the video clip to the desired frame size and center-crop it during pre-training. For downstream tasks, we resize the video clip to frame size or 1.5 times of the frame size, then crop the clip (with frame-size by frame-size spatial size) from the top-left, center, and bottom-right. We apply (horizontal) flip to the raw frames thus a total of 12 spatial augmentations are extracted (12 = 2 resize × 3 crops × 3 flip/no-flip). The only exception is SSV2. This dataset needs to distinguish left/right motions thus we exclude the flip augmentation and only use the center crop during inference following previous works (Feichtenhofer et al., 2019; 2021). During pre-training, we always accumulate the gradient to a batch size of 1024 before updating the weights but use different numbers of negative examples. We analyze this effect in Sec. 5.2.3.

Following previous works (Feichtenhofer et al., 2019), we increase the training epochs for the non-pre-training models by 4× for small datasets (Diving48, UCF101, HMDB51) and 1.5× for larger datasets (SSV2, Kinetics-400).

When analyzing the mask-then-predict task in Sec. 5.2.2 (and all other analysis focusing on mask-then-predict), we exclude the contrastive learning loss (by setting loss weight $\alpha$=0) to preclude potential side effects. However, we still use masked prediction loss when assessing the contrastive learning task in Sec. 5.2.3 as we observe very low performance with only contrastive learning objective.

**Pre-Training with Kinetics datasets.**    Besides pre-training on the HowTo100M (Miech et al., 2019) dataset, we have an experiment with smaller models (i.e., base model in Table 10) pre-trained on K600 dataset. We found that the pure mask-prediction task on K600 (Carreira et al., 2019) can reach 88% on UCF101, but adding the contrastive learning task does not show significant further improvement (+0.5%∼1% according to the hyperparameters and seeds). This is possibly due to K600 dataset is designed specifically for action recognition on shorter video-clip where the average length of the video is 10 second. This short-range video would limit the success of the masked prediction and contrastive learning combination (as illustrated in Table 6 on HowTo100M dataset, $d_{\max}$ = 10s). Given these observations, we did not run the full model (large model in Table 10) on K600 because of the budget constraint.

## D  DATASET DETAILS

In Table 12, we list the key statistics of the datasets used in our paper. HowTo100M is our pre-training datasets that has long-duration uncurated videos. The videos are collected from YouTube by searching key phrases thus the scale could be easily increased. SSV2 and Kinetics-400 are two large downstream datasets, where SSV2 focuses more on the actions and Kinetics-400 focuses more

Table 12: **Key statistics of video datasets** used in this paper. HowTo100M is used for pre-training while others are downstream datasets. The number of training/validation examples in HMDB51 and UCF101 are reported for the train-val split 1.

|  | HowTo100M | SSV2 | Diving48 | UCF101 | HMDB51 | Kinetics-400 |
|---|---|---|---|---|---|---|
| Training | 1238791 | 168913 | 15027 | 9537 | 3570 | 205418 |
| Validation | - | 24777 | 1970 | 3783 | 1530 | 17686 |
| Number of Classes | - | 174 | 48 | 101 | 51 | 400 |
| Average Video Duration | 6.5min | 4s | 6s | 7s | 4s | 10s |

Table 13: Results of **different attention-module layouts and layer-normalization positions**. 'Speed' is the normalized pre-training speed (i.e., number of samples / GPU / second). Models are pre-trained on HowTo100M for 10 epochs. The result numbers represent UCF101 accuracy.

|  | Params | Speed | Pre-LayerNorm | Post-LayerNorm |
|---|---|---|---|---|
| TxHxW | 23.1M | 12.6 | - | 65.9 |
| T,HxW (Bertasius et al., 2021) | 27.1M | 20.0 | - | 69.0 |
| T,H,W | 35.8M | 26.4 | 69.0 | 69.6 |
| T,H—W (ours) | 29.4M | 32.0 | 67.6 | 69.4 |

on the scenes. Diving48, UCF101, HMDB51 are three small datasets. Different from previous datasets on classifying different action types (thus might be potentially inferred from single frames), Diving48 studies the three stages (takeoff, flight, and entry) of competitive diving. Thus achieving good results on Diving48 requires an understanding of the whole video clip. The license of each dataset allows academic use. SSV2 uses the 'Free Academic License'. HMDB51 is licensed under 'Creative Commons Attribution 4.0 International License'. 'The kinetics dataset is licensed by Google Inc. under a Creative Commons Attribution 4.0 International License.' We use the YouTube videos under the Fair Use.

## E    COMPUTATIONAL COST

The pre-training takes around 8.9K V100 GPU hours. This computational cost is at the same level as ViT (Dosovitskiy et al., 2021) supervised training (5.5K hours on ImageNet-21K (Russakovsky et al., 2015) and 12.8K on JFT(Sun et al., 2017)). It is also at the same level of supervised training a model on Kinetics-400 dataset (6.4K for SlowFast (Feichtenhofer et al., 2019), about 5.6K for TimeSformer-L (Bertasius et al., 2021)). For fine-tuning, SSV2, Diving48, UCF101, HMDB51, and Kinetics-400 take 1K, 200, 150, 40, 2K GPU hours, respectively. For analysis, the pre-training takes about 160 GPU hours. Besides the final model training, energy is also spent on tuning the model and finding the best configuration. As shown in Sec. 5.2.3, our method is more robust to the hyperparameters.

## F    ADDITIONAL ANALYSIS RESULTS

### F.1    MODEL ARCHITECTURE COMPARISONS

**Attention Layouts.**    We here compare different alternative model architectures in Table 13. We first experiment with different attention layouts discussed in Sec. 3.4. We consider the sparse attention as proposed in (Child et al., 2019) and the sequential attention blocks as in (Bertasius et al., 2021). The 'TxHxW' model is the basic attention module that takes the flattened tokens as input (of shape T ×H ×W). At each layer, each token attends to all other tokens. The 'T,HxW' model separates the temporal attention and spatial attention ('Divided Space-Time' in (Bertasius et al., 2021)). The 'T,H,W' model processes three attention sequentially ('Axial Attention' in (Bertasius et al., 2021)). The 'T, H—W' model is our default model that sequentially conduct temporal attention and spatial attention, where the spatial attention are parallel into the height attention and width attention. As shown in Table 13, 'T,H—W' reaches a balance between speed and accuracy.

Table 14: Impact of **input resolutions** $T$ **and** $S$. 'Mask-Accu.' and 'CL-Loss' are the pre-training metrics. 'UCF101' indicates the UCF101 fine-tuning results with the pre-training resolution. 'UCF101-Full-Reso.' indicates the full-resolution fine-tuning with $T$=10 and $S$=256.

| #frames $T$ | Frame Size $S$ | Params | Pre-train Speed | Mask-Accu.↑ | CL-Loss↓ | UCF101↑ | UCF101-Full-Reso.↑ |
|---|---|---|---|---|---|---|---|
| 5 | 128 | 29.4M | 32.0 | 17.2 | 1.06 | 69.4 | 73.8 |
| 10 | 128 | 29.4M | 16.5 | 17.2 | 0.96 | 74.2 | 74.6 |
| 5 | 256 | 29.4M | 8.4 | 10.8 | 0.93 | 72.9 | 75.7 |
| 10 | 256 | 29.4M | 4.4 | 10.6 | 0.85 | 78.1 | 78.1 |

Table 15: Impact of **masking ratio**. All models are pre-trained with only mask-then-predict task. #Blocks is the number of masking blocks. Default setup is underlined.

| Strategy | #Blocks | Ratio | Mask-Accu.↑ | UCF101 ↑ |
|---|---|---|---|---|
| Block | 4 | 11.9% | 17.9 | 66.8 |
| Block | 5 | 14.5% | 17.6 | 68.3 |
| Block | 6 | 17.0% | 17.3 | 67.3 |
| i.i.d. | - | 11.9% | 25.6 | 64.5 |
| i.i.d. | - | 14.5% | 24.3 | 63.5 |
| i.i.d. | - | 17.0% | 24.0 | 64.0 |

**Pre-Layer-Normalization vs. Post-Layer-Normalization.** Besides the architectures listed above, we also consider the pre-layer-norm (used in GPT and ViT) and post-layer-norm (used in BERT) variation. We empirically find that post-layer-norm architecture is better for our pre-training tasks as shown in Table 13 (comparing the last 2 columns).

## F.2 INPUT RESOLUTION

In Table 14, we show model scalability over input resolution (i.e., #frames $T$ and frame size $S$). With the same frame size $S$, longer clips perform better than shorter clips (e.g., $T$=10, $S$=128 is better than $T$=5, $S$=128). With the same number of input frames $T$, larger frame size improves the performance (e.g., $T$=10, $S$=256 is better than $T$=10, $S$=128). For each pre-training resolution, we also try to fine-tune under a full-resolution with $T$=10, $S$=256 (denoted as 'UCF101-Full-Reso.'). As in pre-training, fine-tuning with larger resolution generally improves the results. Although longer and smaller clips ($T$=10, $S$=128) show better results than shorter and larger clips ($T$=5, $S$=256) when using the same pre-training and fine-tuning resolutions, they show different trends with the full-resolution fine-tuning. Increasing frame size during fine-tuning (the second block in Table 14) only improves the UCF101 result by 0.4, while increasing the clip length (the third block) improves the UCF101 result by 3.8. These results call for a need of pre-training with large spatial size, and we follow this practice in our large-scale experiments as in Sec. 4.3.

## F.3 NOISY MASKING FOR MASK-THEN-PREDICT

Our default masking strategy replaces all masked tokens with a special [MASK] symbol. We also experiment with BERT's masking strategy that only replaces 80% of masked tokens to the MASK symbol. For other tokens, 10% are randomly sampled from the 'vocabulary' and 10% are kept the same. For smaller experiments, the two masking strategies show similar results. However, this BERT's noisy masking strategy has lower convergence stability on the larger model pre-training. The pre-training diverges after about 10 epochs (out of the 100 epochs).

## F.4 MASKING RATIO FOR BLOCK-MASKING AND I.I.D. MASKING

We test the effect of different masking ratios. In the main text, we control the number of blocks for block masking. In Table 15, we here also show the results of matched masking ratio for i.i.d. masking for completeness. Empirically, the result differences among various masking ratios are marginal and the original BERT's 15% masking ratio (with roughly 5 masking blocks) works slightly

Table 16: Impact of **contrastive learning loss weight** $\alpha$. Default setup is underlined.

| $\alpha$ | Mask-Accu.↑ | CL-Loss↓ | UCF101↑ |
|---|---|---|---|
| 0.0 | 17.6 | - | 68.3 |
| 0.5 | 17.5 | 1.07 | 70.2 |
| 1.0 | 17.2 | 1.06 | 69.4 |
| 2.0 | 16.9 | 1.05 | 68.0 |
| $\infty$* | - | 1.07 | 57.1 |

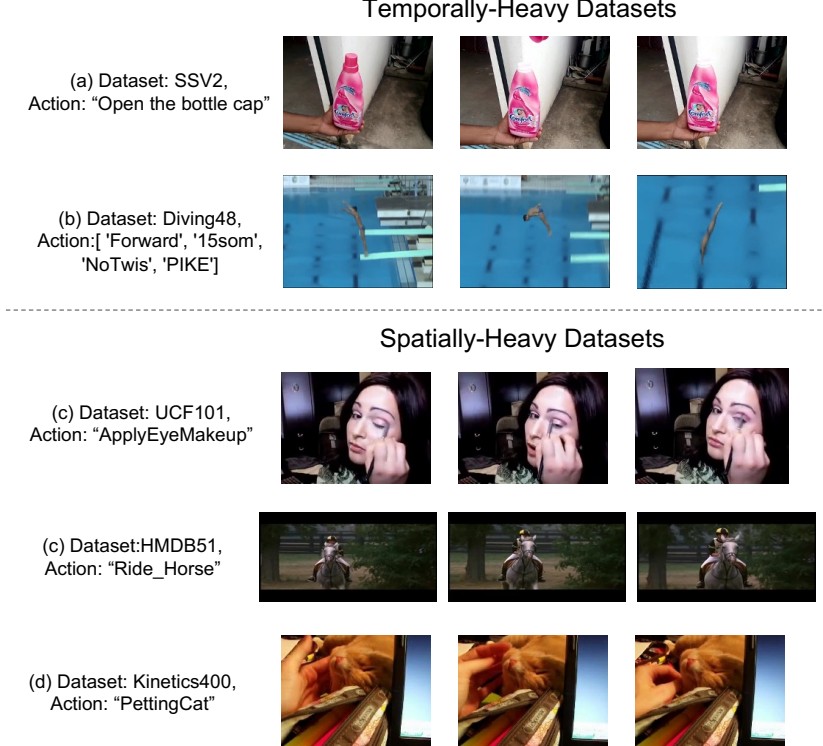

Figure 3: Data samples from temporally-heavy and spatially-heavy datasets. While temporally-heavy datasets need the temporal information to make decisions, most actions in spatially-heavy datasets could be inferred from just a single frame.

better. Thus we always select the number of mask blocks whose induced masking ratio is closest to 15%. For all masking ratios, block masking shows significantly better results than the i.i.d. masking.

## F.5 IMPACT OF CONTRASTIVE LEARNING LOSS WEIGHT

In Table 16, we show the impact of loss weight $\alpha$ (see Sec. 3.3). Since the loss have been calibrated by multiplying the temperature, $\alpha=1$ shows stable results and $\alpha=0.5$ is slightly better. Setting $\alpha=2.0$ will let the model to focus mostly on contrastive learning task and its result is worse than pure mask-then-predict pre-training (i.e., $\alpha=0.0$). We also list the pure contrastive learning pre-training results here (denoted as $\alpha=\infty$* but it excludes the mask-then-predict loss and set $\alpha=1.0$) for reference.

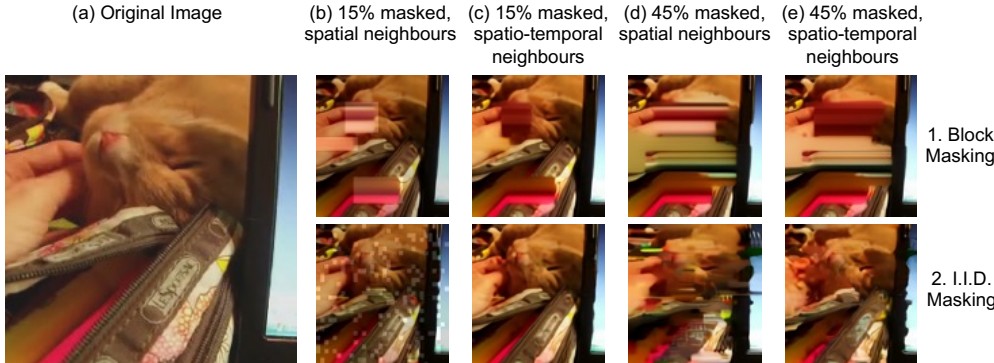

Figure 4: Nearest-neighbour reconstruction of *block masking* and *i.i.d masking*. We mask tokens at different ratios and reconstruct them by simply copying their spatial or spatio-temporal neighbours. Even under heavy masking (e.g., 45% masked), this simple reconstruction strategy still yields a reasonable results for *i.i.d masking*, e.g., we can easily recognize the action 'petting cat' from the reconstructed images, especially the one reconstructed from spatio-temporal neighbours. However, this becomes significantly more difficult when using *block masking*.

## G VISUALIZATIONS

### G.1 TEMPORALLY-HEAVY VS. SPATIALLY-HEAVY DATASETS

We illustrate the differences between temporally-heavy and spatially-heavy datasets in Fig. 3. We here show equally-distributed frames from the video and the label of the video clip. Note that we do not cherry-pick the data but aim for showing the nature of each dataset. Overall, understanding in temporally-heavy datasets needs temporal modeling, whereas the action labels of spatially-heavy datasets could be inferred from a single frame. To understand the SSV2(Goyal et al., 2017) example in Fig. 3.(a), the model needs to understand the causality, i.e., the order of the frames decides the action label. In Fig. 3.(b), the competitive diving dataset Diving48 (Li et al., 2018) also requires considering nearly all frames to make the decision. However, for the spatially-heavy datasets (UCF101 (Soomro et al., 2012b), HMDB51 (Kuehne et al., 2011b), Kinetics-400 (Carreira & Zisserman, 2017)), the action label could be inferred from any single sampled frame. These observations result in the pretty high frame-level accuracy (i.e., not modeling temporal interactions) in Sec. 4.3.

### G.2 MASKING STRATEGY COMPARISONS

We propose to use block masking (in Sec. 3.1) since i.i.d. masking may lead to trivial solutions for the mask-then-predict task given the strong localities in videos. We illustrate this point in Fig. 4 with a simple copy-paste reconstruction method. Specifically, after masking, we first replace the masked tokens with their *nearest visible neighbours* (i.e., the unmasked token that has the shortest distance in spatial or spatio-temporal domain), and then forward the reconstructed tokens to the VQ-VAE decoder to generate the RGB images. For the default 15% masking ratio, i.i.d. masking is recoverable while block masking causes striped noisy patterns.[5] We also test with the extreme case of masking 45% tokens (in Fig. 4 (d), (e)). The block-masked images are hard to reconstruct, however, some objects in reconstructed images from i.i.d. masking are still recognizable. When comparing images under the same masking strategy, recovered images using spatio-temporal neighbours is better than using only spatial neighbours, especially when comparing the images under 45% i.i.d. masking (i.e., (d).2 and (e).2 in Fig. 4). Overall, these results indicate that using i.i.d. masking in mask-then-predict task has a potential trivial solution by copying the neighbourhood, while block-masking resolves this issue.

---

[5]We highlight the masking region in Fig. 4(b) and show the raw RGB images in other cases.

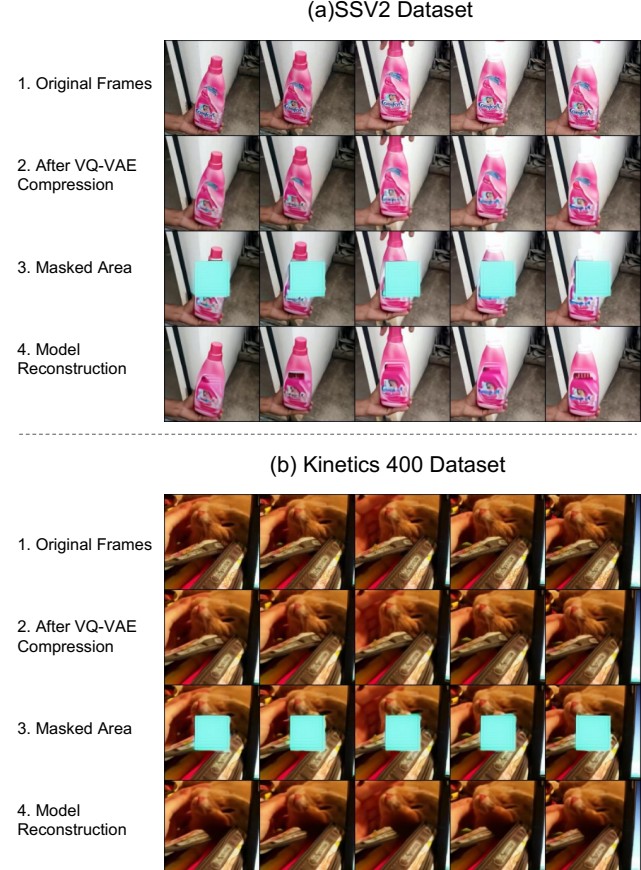

Figure 5: Masked-token model reconstruction for SSV2 and Kinetics-400 datasets. Comparing 1. and 4., our model could redraw temporally-consistent and spatially-plausible patches for the masked regions.

## G.3  MODEL RECONSTRUCTION

Since our model is trained with mask-then-predict task, it is able to reconstruct masked tokens. In this section, we showcase the reconstructed video frames by our final model (i.e., 24 layers, 1024 dimensions, 5 clip length, and 256 frame size). As shown in Fig. 5, we provide two examples from the SSV2 and Kinetics-400 dataset. We uniformly sample 5 consecutive frames from the video at 1 frame per second. We show the original frames in the first rows (Fig. 5.(a).1, (b).1). As illustrated in Sec. G.1, the temporally-heavy SSV2 dataset has object motions between frames while the spatially-heavy Kinetics-400 dataset has almost static frames. In the second rows, we show the images after VQ-VAE compression. To do this, we first use VQ-VAE encoder to encode the images, and then use VQ-VAE decoder to reconstruct the images, without any corruptions in between. We see that there is some information loss caused by the VQ-VAE compression (e.g., the text 'comfort' in Fig. 5.(a).1). It potentially contributes to relative lower results on spatially-heavy datasets. In the third and fourth rows, we illustrate the masked area and the prediction from our model. As shown in Fig. 5.(a).4, our model could faithfully redraw the shape and texture of the object. As shown in Fig. 5.(b).4, the shape of the cat's head is pretty similar to the original frames while the shading is different (but still consistent in different frames).

