# OpenReview forum: "VIMPAC: Video Pre-Training via Masked Token Prediction and Contrastive Learning"
_ICLR.cc/2022/Conference — ICLR 2022 Submitted_

### Official Review · Reviewer_E8qx · 2021-11-01

**Correctness:** 3
**Technical Novelty And Significance:** 2
**Empirical Novelty And Significance:** 2
**Recommendation:** 5
**Confidence:** 4

**Main Review:**

Strength:
+ the paper is well written, main ideas are clear enough, it is also easy to follow for readers.
+ the proposed block masking scheme makes sense in terms of avoiding information leakage from nearby tokens and it is also validated to be quite better than i.i.d.
+ extensive experiments are carried out

Weakness:
- novelty:
（1）Combining masked token prediction (MP) and contrastive learning (CL) is quite straightforward. Both MP and CL are studied previously, simpling combining these techniques is kind of incremental. Yes, the proposed block masking strategy is a plus, but it will have problem by directly combining it to CL. In my opinion, global information loss will be caused by block masking, it is actually harmful to the CL task. If better combination mechanism of the two tasks can be invented, it will be a good work.
  (2) Discrete video token generation is also off-the-shelf technique. VQ-VAE is not new.

- performance and evaluation:
  (1) the performances on "spatially-heavy" datasets, such as UCF101, K400 and HMDB51, are far from the state-of-the-arts, rather than comparable. Such results show that the proposed pretraining method is only good at "local" modeling, its global representation is not strong enough. This may not only be caused by the discretization of VQ-VAE, but also has something to do with the harmful blocking masking which can harm the CL.
 (2)  Now that VQ-VAE can be harmful to the spatial information, why it should still be leveraged. There is also no evaluation on this part. What if VQ-VAE is replaced by other feature encoder and we regress the masked contiguous video feature points? Why not using video frame patches as discrete input tokens?

**Summary Of The Paper:**

This paper proposes a new video pretraining method by combining masked token prediction and contrastive learning. Off-she-shelf VQ-VAE is used in this paper for discrete video tokens generation. In order to make the masked token prediction more effective, the authors proposed a block mask strategy where spatial-temporal neighboring video tokens are masked together. Experimental results are shown to validate its effectiveness.

**Summary Of The Review:**

Given the aforementioned weakness of this paper, I think the novelty of this paper is currently limited, and its performance is not good.

---

> ### Author Response · Authors · 2021-11-22
> **Author Response**
>
> We thank the reviewer E8qx for their comments. We appreciate that they find our method to have extensive experiments and results, well motivated for the block masking design, and be clearly clarified. First, we want to highlight our key contributions/findings: we apply mask-then-prediction pre-training to the important video understanding tasks, combined with contrastive learning. Several techniques (mask blocking, long-term contrastive learning, e.t.c) have been developed  and proven to be essential. The results reach SotA on several temporal understanding tasks. Next, we address the concerns in the review as below.
>
> > Point 1(1): The paper directly combines the block masking strategy to contrastive learning, which is harmful and would have problems.
>
> We disagree with the premise of this statement. “Directly combining two methods might be harmful” is correct, but our paper has specific designs in handling that, and the combined method empirically performs well with our efforts.
>
>  Previous contrastive learning (CL)  focuses on strong spatial augmentation within a short temporal range. This direct CL method will hurt the result (as mentioned in the review) since the low-level semantics are captured by the mask prediction task. Thus, in our paper, we advanced the contrastive learning with long-range sampling and without spatial augmentation. With these critical modifications, the two tasks can now have different focuses: mask prediction helps learn low-level semantics and sequential interaction, while CL helps learn more global and separable representations. This is our effort and novelty in the combination mechanism as the review highlighted.
>
> In another perspective, block masking can be seen as a form of augmentation for CL, as we mask different blocks during training. This gives more motivation to combine the two approaches.
>
> > Point 1(2): Discrete video token generation is also off-the-shelf technique. VQ-VAE is not new.
>
> We do NOT claim VQ-VAE as a novelty in this work as we discussed in the last paragraph of the introduction.
>
> > Point 2(1): The proposed pretraining method is only good at "local" modeling, its global representation is not strong enough?
>
> Not quite, the global understanding of the video is across two dimensions: spatial and temporal, but the review only considers “spatial” as “global”. Our model is good at temporal understanding and we acknowledge the spatial information loss due to VQ-VAE.
>
> Spatial understanding is important but the study of it is covered in the image domain, where temporal understanding is the centric *unique* characteristic of the video research. Thus the improvement over temporal understanding would be important for video research.  We have also shown that “spatially-heavy” dataset is not suitable for assessing model’s temporal understanding, since these datasets can only rely on the single-frame understanding (Table 1, TimeSformer space-only; In CLIP paper, they also show that the single-frame linear-eval approach can reach 92.0 on UCF101, reaching comparative results to other video methods).
>
>
> > Point 2(2): VQ-VAE can lead to a loss of the spatial information, why should it still be leveraged?
>
> Utilizing continuous input is one of our future directions (Sec 4.3). Meanwhile, token-level approach has its distinctive advantages of generative modeling (as shown in Sec G.3, and papers VQ-GAN, DALL-E), more robust to the input noise, and has the possibility to transfer to other modality (e.g., text; since they share the similar types of input and can be modeled using a shared objective). This cross-modality transferability is another future direction that we considered.
>
> Meanwhile, as we discussed in Section 4.3, concurrent work BEiT (Bao et al., 2021) addresses this information loss issue by using image patches instead of VQ-VAE tokens as inputs (but still predict VQ-VAE tokens for masked locations as we do), where they show strong performance on image classification tasks. We encourage future work to study using image patches as inputs for better spatial modeling under our framework.
>
> We have put this discussion in our revision under “Ethical considerations and limitations”.

---

> > ### Comment · Reviewer_E8qx · 2021-11-26
> > **Reply to author reponse**
> >
> >   I am now convinced that the design of block masking + contrastive learning proposed in this paper is potentially quite important.
> > The tokenization process of VQ-VAE can lead to spatial information loss, which significantly degrades the performance on "spatial-heavy" datasets. Actually, if global video representations are well obtained,  the performance on both "spatial heavy" and "temporal heavy" datasets should be consistently improved. The proposed self-training framework performs much worse on ``spatial heavy'' datasets, such a outcome greatly degrades its effectiveness, even though performances on "temporal heavy" datasets are improved.

---

> > > ### Author Response · Authors · 2021-11-30
> > > **Author Response**
> > >
> > > Thanks to the reviewer for their follow-up comments. We are glad that the reviewer is now convinced that the design of our methods is important, and recognize the improvement on "temporal heavy" datasets. Next, we clarify the concerns in the review as below.
> > >
> > > 1. **Performs much worse on "spatial heavy" datasets?** Not quite, our method is comparable on spatially-heavy dataset to other semi-supervised learning from similar uncurated (lower quality, easier to acquire, better scalability) datasets. E.g., on UCF101 (a spatially-heavy dataset), an advanced version of MoCo on IG-Uncurated reached 92.9% and the MIL-NCE trained on HowTo100M reached 91.3%. Our results 92.7% is comparable to their paper, with **+1.4%** to MIL-NCE and **-0.2%** to MoCo. These results can be found in Table 1 of our main paper.
> > >
> > > 2. **A method is not useful if temporal understanding is improved but the spatial understanding is not?** We do not think so. We want to emphasize that previous self-supervised training performs much worser on temporally-heavy dataset. E.g., on SSV2, our VIMPAC method gets 68.1% while previous BYOL and MoCo gets 55.7% (**+12.4%**) and 53.2% (**+14.9%**) respectively. These difference is much significant than what the difference on spatially datasets is, and expose the strong need to advance video's temporal understanding. Since spatial understanding is densely studied in the image domain and the temporal dimension is unique to video data, we think that our method makes a step in the research field by significantly improving the temporal understanding.

---

> > > > ### Comment · Reviewer_E8qx · 2021-11-30
> > > > **Final Rating**
> > > >
> > > > I have read further information provided by the authors and the reviews from other reviewers.
> > > >
> > > > I think this paper is novel to some extent, but the performance on UCF101 is comparable is not so convincing, because the dataset is quite small. The proposed framework performs much worse on Kinetics compared to existing competitors.
> > > >
> > > > Besides, I not mean to say that "a method is not useful if temporal understanding is improved but the spatial understanding is not". Actually, both spatial and temporal information is important for understanding videos. If a model is only good at temporal understanding, it may be a biased one. In fact, the video backbone used in this paper should be capable to model both spatial and temporal dynamics, so it is reasonable to think that the temporal-heavy property can be resulted from the biased pre-training strategy. From this point of view, the proposed pre-training method is also not good enough.
> > > >
> > > > Finally, I would recommend a "marginally below the acceptance threshold".

---

### Official Review · Reviewer_PC2U · 2021-11-02

**Correctness:** 3
**Technical Novelty And Significance:** 2
**Empirical Novelty And Significance:** 2
**Recommendation:** 5
**Confidence:** 4

**Main Review:**

## **Strengths**
* VIMPAC achieves strong empirical results on temporally-heavy datasets.
* Although the mask token prediction task has been proven to be an effective pretraining objective for image tasks, the block masking method designed by authors is well motivated.
* Tokenizing all videos without spatial augmentation is a good way to reduce storage and io cost.

## **Weaknesses**
* This paper is utilizing a transformer as the main architecture, with a pretrained VQ-VAE as the tokenizer. I think it's great to have the baselines with the same arch for fair comparison. Bert-large is a relatively large model comparing to previous CNN models, and large models tend to work better in pretraining with big datasets.
* The contrastive learning objective part doesn't make a lot of sense to me. It seems adding contrastive learning objective only provides marginal improvement, and might even hurt the performance on temporally-heavy datasets. Ablation studies on contrastive learning also lead to some counter-intuitive conclusions so I would expect more analysis and insights from this part.
* I'm also interested if VIMPAC can work for tasks beyond action classification (might be hard due to the architecture design).

**Summary Of The Paper:**

This paper proposed a new pre-training method named VIMPAC for video understanding, which is a combination of specially-designed masked token prediction objective and contrastive learning objective. VIMPAC is well-motivated and shows strong empirical results on several video understanding tasks.

**Summary Of The Review:**

With strengths and weaknesses I listed in previous section, I think this paper makes some reasonable contributions but is marginally below the iclr acceptance threshold.

---

> ### Author Response · Authors · 2021-11-22
> **Author Response**
>
> We thank the reviewer PC2U for their comments. We are glad that they find our method to achieve good results, well motivated for the block masking design, and have advantage in reducing costs. First, we want to highlight our key contributions/findings: we apply mask-then-prediction pre-training to the important video understanding tasks, combined with contrastive learning. Several techniques (mask blocking, long-term contrastive learning, e.t.c) have been developed  and proven to be essential. The results reach SotA on several temporal understanding tasks. Next, we address the reviewer’s concerns below and have incorporated all feedback in our revision.
>
> > Point 1: Why using a BERT-like transformer instead of CNN models?
>
> In this paper, we take discrete tokens as input thus the transformer is a better modeling and convolutional neural network seldomly shows a success on such input. So we focus on understanding the behavior of transformers on this specific input and datasets. Meanwhile, the Transformer model is larger in its number of parameters but does not significantly increase the computational cost. As shown in ViT (Dosovitskiy et.al 2021), the training FLOPs of BERT-Large-like models is comparable to modern convolution-based architectures, even with lower latency.
>
> > Point 2: Does contrastive learning objective only provide marginal improvement?
>
> Not quite, this question was answered in Footnote 4. It is because we kept the small model during the ablation studies due to limited budget. However, contrastive learning (CL) usually needs larger models to show the performance (as in MoCo, SimCLR, etc).
>
> We have experimented with the Base model (but also decreased the training iterations) where CL gives a 3% improvement on UCF101 and the results of temporally-heavy datasets are not hurt. However, we still did not find a significant improvement on a temporally-heavy dataset with contrastive learning on a larger model. The reason might be that the masked prediction can capture the temporal connections between frames but contrastive learning would not help as we expected. This also calls out that the self-supervised study of temporally-heavy tasks should be moved out of the traditional contrastive learning approach, which is the goal of this paper.
>
> We have extended footnote 4 with the above results in the revision and we hope that it can further resolve the confusion.
>
> > Point 3: VIMPAC beyond action classification task
>
> We show the possibility of VIMPAC can be used to recover masked scenes in Sec G.3. It’s also possible to employ VIMPAC backbone to detection tasks where tokens’ outputs can be viewed as anchors. We have put this discussion in our revision under “Ethical considerations and limitations”.

---

> > ### Comment · Reviewer_PC2U · 2021-11-29
> > **Reply to author response**
> >
> > Thanks for the response.
> >
> > 1. I fully agree that Transformers can have better computation efficiency but my main point is the default setting in this paper is "BERT-large" which is a very large model and much larger than commonly used CNN models like Resnet50/101. The author also mentioned unsupervised learning can benefit more from larger model capacity so I'm not sure whether all comparisons are fair in the tables.
> >
> > 2. Contrastive learning objective is known that can work for spatial-heavy dataset and well-explored by previous work. In my opinion, it shouldn't be included in the main contributions in this work if it isn't really important for the most impressive results in this paper. I think the authors should push the limit of the masked token prediction objective and only leave the contrastive learning as an ablation study.
> >
> > In conclusion, I'll keep my original ratings.

---

> > > ### Author Response · Authors · 2021-11-30
> > > **Author Response**
> > >
> > > We thank the reviewer for reading our response and the follow-up comments. Next, we clarify the concerns in the review as below.
> > >
> > > **The contrastive learning design is unimportant and additive to the paper?**
> > >
> > > We first want to point out that a similar question was asked by Reviewer E8qx in their initial review and our response is convincing to them.
> > >
> > > The contrastive learning (CL) design in our paper is important. Directly merging the two methods would not work, but our final combined method empirically performs well with our efforts (explain below). Thus the way to combine them is a contribution brought by this paper, and we also have analysis in identifying the reasons (which we illustrate in Sec 5.2.3, Table 6, Table 8, and briefly explain below).
> > >
> > > Previous contrastive learning (CL) focuses on strong spatial augmentation within a short temporal range. This direct CL method will hurt the result since the low-level semantics are captured by the mask prediction task. Thus, in our paper, we advanced the contrastive learning with **long-range sampling** and **without spatial augmentation**. With these critical modifications, the two tasks can now have different focuses: mask prediction helps learn low-level semantics and sequential interaction, while CL helps learn more global and separable representations. This is our effort and novelty in the combination mechanism.
> > >
> > > In another perspective, block masking can be seen as a form of augmentation for CL, as we mask different blocks during training. This gives more motivation to combine the two approaches naturally.
> > >
> > > For these reasons, we do not agree that our contrastive learning is only additive to the masked prediction method and can be left as an ablation study.  Meanwhile, we thank the reviewer's suggestion, and we would definitely work further on the mask prediction method, trying to push forward the research progress.

---

### Official Review · Reviewer_hrwa · 2021-11-03

**Correctness:** 3
**Technical Novelty And Significance:** 3
**Empirical Novelty And Significance:** 2
**Recommendation:** 5
**Confidence:** 5

**Main Review:**

## Strengths

1. The proposed model is a fresh approach to video modeling and performs well for temporal reasoning tasks like SSv2. The proposed pre-training methods leverage ideas from BERT and shows it can be effective for computer vision tasks too (similar to BEIT).

2. The ablations are interesting, eg table 1 which shows pre-training is imperative for a model like this and lends significant gains.

## Weaknesses

1. Missing/incorrect details
- The paper in Table 1 compares their method to prior work and notes the pre-training dataset used. Authors report their method only uses "HT" for pretraining. However, the tokenizer is using a pre-trained DALL-E model, the training data for which should also be mentioned here? Given this is a paper about learning from unlabeled data, it is important authors are very explicit about *all* the data being used.

2. Continuing from 1), it is unfortunate that no results in Table 1 are comparable to each other; authors use a different pre-training dataset (HT+DALL-E), model architecture (DALL-E + Transformer) and modalities (video only) which makes it impossible to perform any apples-to-apples comparisons. The standard setting in Feichtenhofer et al (2020) seems to be to perform all comparisons with unlabeled Kinetics dataset as pre-training. It would have been ideal if authors kept at least some of the axes of variation fixed. In absence of that, it is only fair to compare the proposed method to state of the art. In that regard it struggles on spatially heavy datasets, and on temporal datasets it performs better although is comparable to more recent SOTA (eg MVIT -- 68.7 without large scale pretraining or HowTo100M pretraining, though it does use K600 labels).

3. The architecture seems quite constrained as it relies on tokenization which looses spatial information. As authors themselves point out, this is likely the reason it struggles on spatial datasets. However, this also means this limitation would limit the impact of the model as it might be very useful for spatial tasks (video detection/segmentation etc). Hence, even though it might be good at learning representations, it is limited to classification. Using a BEIT style framework (as authors also recommend for future work) would make this paper much stronger in that regard.

4. Missing related work: Authors should cite MVIT and other recent transformer based video models. Also while it is not technically published work, I would encourage the authors to cite and compare with video swin transformer.

**Summary Of The Paper:**

The paper proposes a few new techniques: 1) A new video modeling architecture that uses a pre-trained VQ-VAE to tokenize frames, followed by a transformer encoder that aggregates the features and produces the final action class label. 2) Pre-training such an architecture using self-supervision by a) Masked prediction: Authors mask out blocks of tokens and predict them using the context (akin to BERT) and b) contrastive learning: Authors use the representation for two clips from same video as a positive match and otherwise negative match. The final model is trained with a linear combination of the masked prediction and contrastive losses, and finally finetuned for downstream tasks. The model is pretrained on HowTo100M dataset, and finetuned on multiple downstream datasets, where it obtains gains on more temporal datasets like SS-v2.

**Summary Of The Review:**

The proposed model is interesting and in-line with recent work like BEIT etc, and obtains some decent results. However I find it hard to draw meaningful conclusions from the paper due to lack of proper apples-to-apples comparisons. Moreover, given the architecture is likely to have limited impact in the field (as I discuss in my weaknesses), I am borderline on this paper.

---

> ### Author Response · Authors · 2021-11-22
> **Author Response (1/2)**
>
> We thank the reviewer hrwa for their comments. In particular, we are encouraged that they find our method to be fresh, performs well on temporal understanding tasks, and comprehensive ablation studies with interesting findings.   First, we want to highlight our key contributions/findings: we apply mask-then-prediction pre-training to the important video understanding tasks, combined with contrastive learning. Several techniques (mask blocking, long-term contrastive learning, e.t.c) have been developed  and proven to be essential. The results reach SotA on several temporal understanding tasks. Next, we address the reviewer’s concerns below and have incorporated all feedback in our revision.
>
> > Point 1: The training data for VQ-VAE should also be mentioned here?
>
> Thanks! The VQ-VAE training data is described in Sec A.1 of the paper. We have updated the discussion in Sec. 4.1 Datasets in our revision.
>
> However, we did NOT think that VQ-VAE helps with a better representation for the video pre-training, but rather consider it as a compression method (Sec A.1; Page 14 bottom). Reasons as below: (1). The VQ-VAE encoder largely compresses a 8x8x3 vector (ranging from 0-255) to an integer of 0-8191. (2). Without pre-training (in Table 4 row1), models on VQ-VAE codes show much lower results than from the raw pixels. Previous work (TimeSformer [5]) using raw pixels with a similar architecture as ours gets 62.3% on SSV2, without pre-training, our model failed to converge, and gets only 1.2%. It suggests that VQ-VAE might not provide a “stronger representation” for the model to learn. (3) We did not see clear evidence from previous publications that VQ-VAE creates a “strong representation” for visual recognition tasks. (4) Increasing model size (shown in Table 2) has a strong impact on performances, which shows that this is not only VQ-VAE representation related. For these two reasons, we excluded it from the list of pre-training datasets but we made explicit clarification in the main text of the updated version.
>
> Meanwhile, the VQ-VAE compression can be also trained on the HT video dataset solely, which are in-domain and likely further boost the performance. We select the DALL-E pre-trained snapshot to be fairly comparable to other works and can be potentially easier transferred to other domains. In a similar fashion, we note that BEiT (Bao et al 2021) does not mention DALL-E’s training data as well when comparing to other methods.
>
> > Point 2: Why not use pre-Training results on K600?
>
> We had preliminary results on K600 (discussed below and added to the revision). But K600 videos are short in range thus it is not suitable for our long-range CL methods. We thus exclude the full experiments due to limited budget.
>
> We found that the long-range sampled contrastive learning is essential for combining mask prediction (MP) task and contrastive learning (CL) task. In our design, the two tasks have different targets: the mask prediction helps local representation and the contrastive learning helps global understanding. If short-range sampling is used, the two tasks will all learn towards local representations and the result drops a lot (in Table 6).
>
> Unfortunately, K600 dataset is designed specifically for action recognition on video-clip where the average length of the video is *10 second* (HT is *6 mins*). We have an experiment with smaller models (Base Model in Table 10) on K600. We found that K600+MP can reach 88 on UCF101, but adding the CL task does not show significant further improvement (+0.5~+1 according to the hyperparameters and seeds). This is possibly due to the short range of K600 videos (as illustrated on HT in Table 6, d_max = 10s).  At the same time, although KX00 is smaller in datasize, models are running longer to achieve good results (800 epochs in Feichtenhofer et al 2021 and Qian et al 2020, e.t.c) thus the computation cost is still high. Given these observations, we have not run the full model (Large Model in Table 10) on K600 because of the budget constraint. Since this smaller model on K600 is not comparable with the final model, we exclude it from our initial submission.
>
> We have added these results and discussions in Sec. Appen.C of our revision.
>
>
> > Point 2: Better results from the supervised methods (e.g., MVIT)?
>
> Yes. However, as already mentioned in the review, the method is supervised pre-trained and thus not compared to our self-supervised methods. Thanks for the pointer and we have updated the discussions in the revision.

---

> > ### Author Response · Authors · 2021-11-22
> > **Author Response (2/2)**
> >
> > > Point 3: Token-based approach would limit the impact of the model
> >
> > We are among the first line of work that use masked token reconstruction, etc. as a self-supervised learning approach for vision, and we do agree the use of VQ-VAE tokens might limit the models' ability in certain aspects, such as spatial info loss. Future work could explore combining the techniques from BEiT for a more capable approach.
> >
> > Meanwhile, token-level approach has its unique advantage of generative modeling (as shown in Sec G.3, and papers VQ-GAN, DALL-E), more robust to the input noise, and has the possibility to transfer to other modality (e.g., text; since they share the same types of input). This cross-modality transferability is another future direction that we consider.
> >
> > We have updated these discussions in our revision under “Ethical considerations and limitations”.
> >
> > > Point 4: Cite MVIT and other recent transformer
> >
> > Thanks for the references! We have cited the above paper and add the discussions in Sec. 2 (Related Works) of our revision. We also want to highlight that these recent papers focus on the model improvement while our paper focuses on the pre-training methods. Besides the differences of focuses, the hierarchical design (the feature map is gradually reduced) of MVIT and Video Swin Transformers is also not directly applicable to the mask prediction tasks.

---

> > > ### Comment · Reviewer_hrwa · 2021-11-29
> > > **Final rating**
> > >
> > > I thank the authors for the response. I’ve looked through it and other reviews, and plan to stick to my original review/rating. A few additional thoughts:
> > >
> > > 1. “We select the DALL-E pre-trained snapshot to be fairly comparable to other works and can be potentially easier transferred to other domains.”
> > > I’m not sure how DALL-E makes comparisons fairer given no previous video SSL techniques use it. Also if it makes the representation potentially easier to transfer, that means it learns something better than what one could learn from HT only? I think the only sure shot way to show it does not matter is to train the vqvae on HT. Also, I’m not sure why BEIT does not mention the VQVAE training data, but I don’t think that is a valid argument on why it should not be mentioned here. Follow up work like “masked auto encoder” that compare to  BEIT do mention that additional data in the tables.
> > >
> > > 2. “The hierarchical design (the feature map is gradually reduced) of MVIT and Video Swin Transformers is also not directly applicable to the mask prediction tasks.” Not sure why that would be. Please see https://arxiv.org/abs/2111.09886

---

> > > > ### Author Response · Authors · 2021-11-30
> > > > **Author Response**
> > > >
> > > > We thank the reviewer for the follow-up comments and for careful checking our previous response and other reviews. Next, we clarify the concerns in the review as below.
> > > >
> > > > 1. **The hierarchical design is not directly applicable.**
> > > >
> > > >    The inputs, outputs and targets of our method are VQVAE tokens, while for https://arxiv.org/abs/2111.09886, they are raw pixels. For our method, if the tokens got merged in the intermediate layers (in MViT and Video Swin Transformer), there will be a mismatch between the output sequence and the target, e.g., if input sequence 256 (each token corresponds to a 8x8 patch), after some merging, the output sequence length becomes 64, while the target sequence length is still 256. For https://arxiv.org/abs/2111.09886, since it uses raw pixels, after the same merging, it can directly predict patch pixel values corresponds to the merged patch, which is 16x16 (merged from the original four 8x8 input patches).
> > > >
> > > > 2. **The off-the-shelf VQ-VAE training data is part of the video pre-training data?**
> > > >
> > > >     We first want to re-emphasize our point that VQ-VAE **does not help with a better representation** for the video pre-training, but rather consider it as a compression method (Sec A.1; Page 14 bottom). Please see the original response about the four evidence with experimental supports. Thus we did not list it as part of the video pre-training data but add clear discussions in the paper. For another point "re-training the VQ-VAE on HT100M", the OpenAI DALL-E code only releases the model but **not the training scripts** while the paper has several unclear points about the detailed implementation. Multiple issues on GitHub has been post regarding this: https://github.com/openai/DALL-E/issues/8, https://github.com/openai/DALL-E/issues/31, https://github.com/openai/DALL-E/issues/42. We have kept tracking the Github for a long time, and we are definitely happy to run the VQ-VAE experiments on HT100M in the future if the training code is released and the computational budget is allowed.
> > > >
> > > > 3. **Regarding recent works MAE and SimMIM.**
> > > >
> > > >      We thank the reviewer for bringing these two papers into discussion. These papers both firstly appear on ArXiV on **Nov 2021**. We are glad to incorporate the discussions about them into our paper, but we want to emphasize that they are both strictly later works.

---

### Decision · Program_Chairs · 2022-01-20

**Decision:**

Reject

**Comment:**

None of the reviewers recommended this paper. There were concerns that it is hard to draw meaningful conclusions from the experimental work due to the comparisons provided.  While the design of the block masking + contrastive learning proposed in this paper was rated as potentially being quite important, there remained some concern that subsequent tokenization steps could be problematic for "spatial heavy" datasets.

The AC recommends rejection.